Wide Discrepancies in the Magnitude and Direction of Modelled SIF in Response to Light
2                                         Conditions

Nicholas C Parazoo[1], Troy Magney[1,2], Alex Norton[3], Brett Raczka[4], Cédric Bacour[5], Fabienne
Maignan[6], Ian Baker[7], Yongguang Zhang[8], Bo Qiu[8], Mingjie Shi[9], Natasha MacBean[10], Dave R.
Bowling[4], Sean P. Burns[11,12], Peter D. Blanken[11], Jochen Stutz[9], Katja Grossman[13], Christian
7                                         Frankenberg[1,2]

Jet Propulsion Laboratory, California Institute of Technology[1]
California Institute of Technology[2]
School of Earth Sciences, University of Melbourne[3]
School of Biological Sciences, University of Utah[4]
NOVELTIS, 153 rue du Lac, 31670 Labège, France[5]
Laboratoire des Sciences du Climat et de l'Environnement, LSCE/IPSL[6]
Colorado State University[7]
International Institute for Earth System Sciences, Nanjing University, China[8]
University of California Los Angeles[9]
Department of Geography, Indiana University[10]
Department of Geography, University of Colorado[11]
National Center for Atmospheric Research[12]
Institute of Environmental Physics, University of Heidelberg[13]

Prepared for Biogeosciences

**Abstract:**

Recent successes in passive remote sensing of far-red solar induced chlorophyll fluorescence (SIF) have spurred development and integration of canopy-level fluorescence models in global terrestrial biosphere models (TBMs) for climate and carbon cycle research. The interaction of fluorescence with photochemistry at the leaf- and canopy- scale provides opportunities to diagnose and constrain model simulations of photosynthesis and related processes, through direct comparison to and assimilation of tower, airborne, and satellite data. TBMs describe key processes related to absorption of sunlight, leaf-level fluorescence emission, scattering and reabsorption throughout the canopy. Here, we analyze simulations from an ensemble of process-based TBM-SIF models (SiB3, SiB4, CLM4.5, CLM5.0, BETHY, ORCHIDEE, BEPS) and the SCOPE canopy radiation and vegetation model at a subalpine evergreen needleleaf forest near Niwot Ridge, Colorado. These models are forced with local meteorology and analyzed against tower-based continuous far-red SIF and gross primary productivity (GPP) partitioned eddy covariance data at diurnal and synoptic scales during the growing season (July-August 2017). Our primary objective is to summarize the site-level state of the art in TBM-SIF modeling over a relatively short time period (summer) when light, canopy structure, and pigments are similar, setting the stage for regional- to global-scale analyses. We find that these models are generally well constrained in simulating photosynthetic yield, but show strongly divergent patterns in the simulation of absorbed photosynthetic active radiation (PAR), absolute GPP and fluorescence, quantum yields, and light response at leaf and canopy scale. This study highlights the need for mechanistic modeling of non-photochemical quenching in stressed and unstressed environments, and improved representation of light absorption (APAR), distribution of light across sunlit and shaded leaves, and radiative transfer from leaf to canopy scale.

**Section 1: Introduction**

Our ability to estimate and measure photosynthesis beyond the leaf scale is extremely limited. This inhibits the ability to evaluate the performance of terrestrial biosphere models (TBMs) that are designed to quantify the direct impact and feedbacks of the carbon cycle with climate change. Consequently, there are substantial uncertainties in estimating the gross primary production (GPP) response to environmental changes and carbon-climate feedback (Friedlingstein et al., 2014). Global, multi-scale remote sensing of solar induced fluorescence (SIF) may represent a major breakthrough in alleviating this deficiency (Mohammed et al, 2019). Spaceborne data indicate a linear relationship between SIF and GPP at large spatial (kilometer) and temporal (bi-weekly) scales (e.g., Sun et al., 2017) for several ecosystems, while theoretical models and ground-based measurements indicate a more non-linear relationship at leaf and canopy scales (Zhang et al., 2016; Gu et al., 2019; van der Tol et al., 2014; Magney et al., 2017, 2019a).

Chlorophyll fluorescence is re-emitted energy produced during the photosynthetic light reactions, in which a small fraction (roughly 2%) of photosynthetic active radiation (PAR) absorbed by chlorophyll is re-emitted at longer wavelengths (650-850 nm) as fluorescence. In ambient conditions, the emission of SIF represents a by-product of two primary de-excitation pathways, photochemical and nonphotochemical quenching (PQ, NPQ). Plants have evolved these regulatory mechanisms to prevent damage to photosynthetic machinery when the amount of absorbed radiation is greater than that which can be used to drive photochemistry. Chlorophyll fluorescence responds dynamically to changes in photochemistry and NPQ from instantaneous to hourly, daily, and seasonal timescales, as a function of changing environmental conditions and plant structural properties (Porcar-Castell et al., 2014; Demmig-Adams et al., 2012). SIF is fundamentally different than steady-state fluorescence yield typically measured at the leaf scale as it is sensitive to both changes in photochemistry as well as absorbed PAR (APAR, related to incident light, canopy structure, and biochemical content). The response of canopy SIF to APAR is well documented in deciduous and evergreen forests and cropping ecosystems (Yang et al., 2018; Badgley et al, 2017; Miao et al., 2018; Magney et al., 2019b; Li et al., 2020). More recently, Magney et al. (2019b) showed that seasonal changes in canopy SIF for cold climate evergreen

systems is influenced by changes in needle physiology and photoprotective pigments (Magney et
al., 2019b).
To properly account for these factors, process-based SIF models must represent these underlying
non-linear biophysical and chemical processes. Several modeling groups have adapted TBMs to
incorporate various SIF formalisms for the purpose of model evaluation, data assimilation, and
improved model prediction (Lee et al., 2015; Koffi et al., 2015; Thum et al., 2017; Norton et al.,
2019; Bacour et al., 2019; Raczka et al., 2019). With these goals in mind, TBM SIF modeling
requires two important steps: (1) a representation of SIF at the leaf scale that accounts for NPQ
and photochemistry, and (2) canopy radiative transfer of SIF, which enables a comparison to large
field-of-view observations (e.g. tower, satellites). The second step involves accounting for
radiative transfer within the canopy and has typically relied on incorporating the Soil Canopy
Observation Photosynthesis Energy model (SCOPE, van der Tol et al., 2009, 2014), which
simulates chlorophyll fluorescence as a function of biophysics, canopy structure, environmental
conditions, and sun/sensor geometries. This approach has been adopted by TBMs in various ways
using different assumptions for fluorescence modeling and radiative transfer, as will be discussed
in Section 2.
Typically, measuring chlorophyll fluorescence and competing pathways (PQ, NPQ) has been done
at the leaf scale via pulse-amplitude modulation fluorescence (PAM, Schreiber et al., 1986).
Recently, commercially available spectrometers have made it possible to measure SIF directly in
the field at the leaf and canopy scale, and also enable the study of structural, environmental, and
directional controls (Cogliati et al. 2015; Daumard et al. 2010; Migliavacca et al. 2017; Yang et al.
2015; Grossman et al., 2018; Aasen et al., 2019; Gu et al., 2019b; Zhang et al., 2019). The use of
field deployable instruments on eddy covariance towers has increased rapidly since 2014,
providing coverage of multiple vegetation types across various climates around the world (Yang
et al., 2018; Magney et al., 2019a,b; Parazoo et al., 2019). These data enable improved
understanding of the relationship between SIF, GPP, APAR, and environmental effects at canopy
scales. Novel tower-mounted spectrometer systems such as Fluospec2 (Yang et al., 2018),
Photospec (Grossman et al., 2018), and FLOX (e.g., Julitta et al., 2017; Shan et al., 2019) have
made it possible to monitor canopy SIF continuously in the field with high precision over multiple
years providing opportunities for more direct comparison and evaluation of satellite data
(Grossman et al., 2018; Yang et al., 2015, 2018; Wohlfahrt et al., 2018; Magney et al., 2019b).
PhotoSpec offers the additional benefits (and challenge) of (a) precise field of view capable of
resolving leaf-level SIF, and (b) canopy scanning at azimuth and elevation angles. These features
enable SIF integration from leaf- to canopy- scales, and interpretation of directional variations of
the emitted radiance.
Canopy scanning spectrometers such as PhotoSpec thus provide an opportunity to understand
the physical processes that lead to a breakdown of SIF-GPP linearity at leaf to canopy scale (or
conversely, emergence of linearity at increasing scale), and for detailed evaluation and diagnosis
of TBM performance. This study provides a preliminary benchmarking site-level assessment for
simulations of SIF within a TBM framework and across an ensemble of TBMs, with the primary
purpose being an initial investigation into the response of modelled SIF and GPP to light during
peak summer. We leverage continuous measurements of SIF and GPP at the Niwot Ridge US-NR1
Ameriflux flux tower in Colorado from June-July 2017 (Magney et al., 2019b), and simulations of
canopy radiative transfer, photosynthesis, and fluorescence from a stand-alone version of SCOPE,
to (1) Benchmark TBM-SIF modeling, (2) Evaluate sensitivity to underlying processes and scaling
techniques, (3) Identify strengths and weaknesses in current modeling strategies, and (4)
Recommend strategies for models and observations.
The paper is organized as follows: Section 2 describes SCOPE and the seven TBM-SIF models (SiB3,
SiB4, ORCHIDEE, BEPS, BETHY, CLM4.5, CLM5) which have recently been published or are in
review, and provides more details on site level benchmarking observations. Section 3 summarizes
results comparing modelled and predicted SIF and GPP at hourly and daily scales, as they relate
to absorbed light, GPP and SIF yields, and quantum yields. Section 4 discusses results in more
detail, including attribution of SIF magnitude and temporal phasing biases and sensitivities to
absorbed light, and areas for improvement.
**Section 2: Methods**
*2.1 Site: Niwot Ridge, Colorado*
Our study focuses on an AmeriFlux (https://ameriflux.lbl.gov/) site in Niwot Ridge, Colorado,
USA (US-NR1), where a tower-based eddy covariance system has been continuously measuring
the net ecosystem exchange of carbon dioxide (NEE) over a high-elevation subalpine forest
since 1999, and a spectrometer system that has been continuously monitoring SIF since June
2017 (Grossman et al., 2018; Magney et al., 2019b). The 26 m tall tower is located in a high
elevation forest (3050 m asl) located in the Rocky Mountains of Colorado (Burns et al., 2015; Hu
et al., 2010; Monson et al., 2002) and consists primarily of the evergreen species of lodgepole
pine (*Pinus contorta*), Engelmann spruce (*Piceae engelmanii*), and subalpine fir (*Abies*
*lasiocarpa*). The mean annual temperature is 1.5°C and mean annual precipitation is 800 mm
(65% as snow). The forest is roughly 120 years old with a mean canopy height of 11.5 m, and a
leaf area index of 4.2 $m^2$ $m^{-2}$. More site-specific details can be found in Burns et al. (2015).
At Niwot Ridge, interannual variations in GPP are closely linked to winter snowfall amount, which
typically melts by early June, and summer precipitation, characterized by afternoon convective
thunderstorms triggered by upslope flow (Burns et al., 2015; Albert et al., 2017) and
climatological peak precipitation around 2 pm local time (Fig 1A). We note that our study period
of July-August 2017 is unusual for NR1 (relative to the 2015-2018 mean) in its bimodal
distribution of diurnal precipitation (morning and afternoon peaks), lower than normal afternoon
precipitation, cooler temperatures, and reduced vapor pressure deficit (Fig 1 A-C). The early
morning peak is due to a strong storm system that moved through from July 22-24 (Fig 1E), and
does not show up when these days are removed. This period also shows a decrease in incoming
shortwave relative to climatology despite lower precipitation (Fig 1D). We note that a second
storm passed through in early August. The combination of these two storms produced net
decreases in air temperature (Fig 1F), vapor pressure deficit (Fig 1G) and sunlight (Fig 1H) over a
two-week period from late July to early August.
2.2 Tower-Based Measurements: PAR, SIF, $CO_2$ Flux
2.2.1 Absorbed PAR
The site is equipped with two main upward-facing PAR sensors. The first (LICOR LI-190R),
mounted on the PhotoSpec telescope unit, provides an independent measurement of
direct/diffuse light and can be used to calibrate PhotoSpec (Grossman et al., 2018). The second
(SQ-500-SS; Apogee Instruments), mounted on the main flux tower, is part of a larger array of
upward- and downward-oriented PAR sensors above and below the canopy used for the
calculation of the fraction of PAR absorbed by the vegetation canopy (fAPAR). The two PAR
sensors show a similar diurnal pattern during July-August 2017 (Fig S1), including an afternoon
dip and relatively smaller values overall compared to 2018 (the only other year with available
PAR for comparison).
Full-spectrum quantum sensors (SQ-500-SS; Apogee Instruments) were new and factory-
calibrated together just before installation. Above-canopy sensors (one up and one down-facing)
were mounted on the main flux tower, and below-canopy sensors (six up and six down) were
mounted at the 2 m height above ground on a shorter canopy-access towers. APAR was
calculated for each pair of below-canopy relative to above-canopy sensors for every half-hour,
then averaged among sensors over daylight hours to create a daytime average. We then estimate
hourly APAR by multiplying hourly incoming PAR (measured and integrated from 400-700 nm) at
the top of canopy (PAR) by the daytime average of fAPAR. Fig S2 shows the mean diurnal cycle
for July-August 2017 for each sensor, and the across-sensor average, with APAR data collection
beginning on July 13, 2017. We note that APAR measurements are only as representative as the
distribution of PAR sensors beneath the canopy; while they are placed within the footprint of SIF
(Sec 2.2.3) and fetch of eddy covariance (Sec 2.2.4) measurements, they cannot be a perfect
representation of canopy APAR for each eddy covariance and SIF measurement.
2.2.2 Fluorescence parameters
We define and clarify three important quantities that define the relationship between absorbed
light and emitted SIF at leaf and canopy scales. $\phi_F$ is the quantum yield of fluorescence,
representing the probability an absorbed photon will be fluoresced. This quantity can be
observed at leaf level using PAM fluorimetry or calculated by models as a function of rate
coefficients for energy transfer (Sec 2.3.3). $SIF_{yield}$ is the canopy emitted SIF per photon absorbed.
The quantity is estimated from models and observations as the ratio of absolute canopy SIF and
APAR ($SIF_{canopy}$/APAR). $SIF_{yield}$ is our best attempt to account for the effects of (a) canopy absorbed
light and (b) SIF re-absorption within the canopy on the canopy integrated emission of SIF.
However, factors such as observation angle, fraction of sunlit/shaded canopy components, and
difference in footprint from APAR, necessitates an additional diagnostic variable defined as
relative SIF ($SIF_{rel}$). $SIF_{rel}$ is emitted SIF per reflected radiance in the far red spectrum where SIF
retrievals occur ($SIF/Ref_{fr}$). This is useful because is normalizes for the exact amount of
'illuminated' canopy components within the sensor field of view, whereas APAR measurements
are integrated for the entire canopy.
These quantities represent different but equally important versions of reality. It is difficult for
models to exactly reproduce the distribution and timing of sunlight in the canopy as observed by
PhotoSpec. While $SIF_{rel}$ removes model-observation differences in illumination, it confounds our
interpretation of the relationship with $GPP_{yield}$, which is derived from APAR. As such, we provide
both results to be comprehensive, but note the temporal stability associated with $SIF_{rel}$ as the
more physical interpretation of canopy yield for this short period of study.
2.2.3 Tower Based Measurements of Solar Induced Chlorophyll Fluorescence (SIF)
SIF data has been collected from a scanning spectrometer (PhotoSpec) installed at the AmeriFlux
US-NR1 tall tower since June 17, 2017. PhotoSpec sits atop the tower at 26 m above the ground
and roughly 15 m above the forest canopy top, transferring reflected sunlight and SIF data
collected from the needleleaf canopy through a tri-furcated optical cable to three spectrometers
in a shed at the base of the tower. These spectrometers measure far-red fluorescence in the 745-
758 nm retrieval window at high spectral resolution (FWHM = 0.3 nm) and with a 0.7 deg field of
view (FOV), resulting in a 20 cm diameter footprint at nadir on top of the canopy. The far-red SIF
data are then scaled to 740 nm for model intercomparison using the first principal component of
the spectral shape in Magney et al., 2019a. Photospec scans from nadir to the horizon in 0.7
degrees steps at two azimuth directions, with a time resolution of ~20 s per measurement and
complete scan time of 20 minutes. For this study, we aggregate scans across all azimuth and
elevation angles into hourly, canopy level averages to benchmark model estimates of top of
canopy (TOC) or canopy averaged SIF (BETHY only, see Sec *2.3.4*.1) at diurnal and synoptic time
scales. We refer the reader to Grossman et al. (2018) and Magney et al (2019b) for further details
regarding PhotoSpec, implementation at US-NR1, and data filtering, and to Magney et al (2019c)
for data access. We focus our model-data analysis on the 2017 growing season (July-August,
2017) to maximize overlap between observations of SIF, GPP, and APAR.
Diurnal composites of PhotoSpec SIF in 2017 show a late morning peak and afternoon dip (Fig
S3A). The afternoon dip is consistent with decreased incoming shortwave, PAR and APAR (Figs S1
and S2, respectively). However, we note the retrieved signal from PhotoSpec is also affected by
(1) viewing geometry, (2) fraction of sunlit vs shaded leaves (sun/shade fraction, i.e. the quantity
of needles illuminated by incident sunlight) due to self-shading within the canopy, and (3)
direct/diffuse fraction due to cloud cover. Structural and bidirectional effects lead to different
SIF emission patterns depending on view angle and scanning patterns (Yang and van der Tol,
2018). The viewing geometry of PhotoSpec (as implemented at NR1 in 2017) causes a higher
fraction of illuminated vegetation in the morning, which leads to a 2 to 3 hour offset in the timing
of peak SIF (Fig S3A) and incoming far-red reflected radiance within the retrieval window (Fig
S3B), from the peak zenith angle of the sun at noon (coinciding with the expected peak in PAR)
to late morning. Normalizing SIF by far-red reflected radiance as relative SIF ($SIF_{rel}$, Fig S3C) and
rescaling to SIF (Fig S3D) shifts the peak back to noon and preserved the afternoon dip (albeit
with reduced magnitude). $SIF_{rel}$ helps to account for factors 1-3 listed above because it accounts
for the amount of reflected radiation in the field of view of PhotoSpec, which is impacted by
canopy structure, sun angle, and direct/diffuse light. As discussed above, $SIF_{rel}$ is likely a better
approximation of $SIF_{yield}$ because it normalizes for the exact amount of 'illuminated' canopy
components in each retrieval, whereas APAR integrates the entire canopy. As such, we expect
$SIF_{rel}$ to have a strong seasonal change associated with downregulation of photosynthesis, and a
more subtle diurnal change, as during mid-summer the SIF signal is primarily driven by light
intensity.
It is important to note that the PhotoSpec system is highly sensitive to sun/shade fraction in the
canopy (factor 2) due to the narrow FOV of the PhotoSpec telescoping lens. Increased afternoon
cloud cover during summer causes diurnal asymmetry in incident PAR (Fig S1A). We examine this
effect in more detail (Section 3) by analyzing SIF and GPP under clear and diffuse sky conditions
using a threshold (0.5, top-of-canopy/top-of-atmosphere incoming shortwave radiation) similar
to that used in Yang et al. (2017) and Yang et al. (2018).
2.2.4 $CO_2$ Flux and GPP Partitioning
NEE measurements are screened using $u_{star}$ filtering, and partitioned into gross primary
production (GPP) and terrestrial ecosystem respiration components using the so-called nighttime
method which is based on the relationship between NEE during the nighttime (PAR < 50 umol m$^-$
$^2$ s$^{-1}$) and air temperature (Reichstein et al., 2005). Diurnal averages of GPP based on nighttime
partitioning show similar diurnal structure to PAR and SIF including the afternoon dip and
reduced overall magnitude compared to the 2015-2018 mean (Fig S4). Similar results are found
using daytime light partitioning of NEE (Lasslop et al., 2010; Fig S4) and thus only nighttime
partitioned GPP data are reported for the remainder of this study. All GPP estimates are
processed as half hourly means, then gap filled and averaged hourly. We note the tower location
near the Continental Divide in the Rocky Mountains of Colorado presents slope flow challenges
for eddy covariance during nighttime, but the relatively flat area of the tower reduces impact on
daytime flux measurements (Burns et al., 2018). Details on the flux measurements, data
processing and quality control are provided in Burns et al. (2015).
2.3 Modeling Approach
*2.3.1 TBM-SIF Overview*
The parent TBMs are designed to simulate the exchanges of carbon, water, and energy between
biosphere and atmosphere, from global to local scales depending on inputs from meteorological
forcing, soil texture, and plant functional type. The addition of a fluorescence model that
simulates SIF enables a direct comparison to remotely sensed observations for benchmarking,
process diagnostics, and parameter/state optimization (data fusion) for improved GPP
estimation. The TBM-SIF models analyzed here differ in ways too numerous to discuss. We refer
the reader to the appropriate references in Section 2.3.4 for more detailed model descriptions.
Instead, we focus on key differences affecting joint simulation of GPP and leaf/canopy level SIF
at diurnal and synoptic scale, during the peak of summer. These differences, which are
summarized in Table 1, include the representation of stomatal-conductance (all use Ball-Berry
except CLM5.0, BEPS, and ORCHIDEE), canopy absorption of incoming radiation (all account for
sunlit/shaded radiation except ORCHIDEE, SiB3, and SiB4), limiting factors for photosynthesis
($V_{cmax}$, LAI, radiation, stress) and SIF ($k_N$, fluorescence photon re-absorption), scaling and radiative
transfer methods for transferring leaf-level SIF simulations to top of canopy, and parameter
optimization. Further details on (a) photosynthetic structural formulation and parameter choice,
(b) representation of leaf level processes important to SIF ($k_N$ and $\phi_P$), and (c) leaf-to-canopy
scaling approach ($SIF_{canopy}$) are provided in Sections 2.3.2 and 2.3.3.
*2.3.2 Photosynthesis Models*
All TBM-SIF models in this manuscript used enzyme-kinetic models to simulate leaf assimilation
rate (gross photosynthesis) as limited by the efficiency of photosynthetic enzyme system, the
amount of PAR captured by leaf chlorophyll, and the capacity of leaves to utilize end products of
photosynthesis (Farquhar et al., 1980; Collatz et al., 1991, 1992; Sellers et al., 1996). However,
there are important differences in the representation of (a) stomatal conductance that couples
carbon/water cycles, and (b) limiting factors on carbon assimilation due to leaf physiology
(maximum carboxylation capacity, $V_{cmax}$), radiation (APAR or fAPAR), canopy structure (LAI, leaf
angle distribution), and stress (water supply and demand, temperature), that affect plant
physiological processes and canopy radiative transfer. The underlying stomatal conductance
models in the TBMs analyzed here are represented by the Ball-Berry family of empirical models
rooted in the leaf gas exchange equation but with different representations of atmospheric
demand (relative humidity or vapor pressure deficit), including the Ball-Berry-Woodrow model
(Ball et al., 1987), the Leuning model (Leuning, 1995), the Yin-Stuik model (Yin and Struik, 2009),
and the Medlyn model (Medlyn et al., 2011). These structural and parametric differences also
influence calculated values such as the degree of light saturation (Section 2.3.3), which influence
both the fluorescence and quantum yield as used by the fluorescence models. Differences in
stomatal conductance, canopy type / radiation scheme, stress, $V_{cmax}$, and LAI are summarized in
Table 1.
*2.3.3 Fluorescence Modeling Approach*
Following the general approach described in Lee et al. (2015) and van der Tol et al. (2014), the
flux of total leaf-level emitted fluorescence, $SIF_{leaf}$, can be diagnosed using a light use efficiency
framework analogous to the expression for photosynthesis (Monteith et al., 1972),
$$SIF_{leaf} = fAPAR * PAR * \phi_F$$
$$= APAR * \phi_F \qquad\qquad \textit{Equation 1}$$
where *PAR* and *fAPAR* are defined in Section 2.2.1 but measured at leaf level, and $\phi_F$ is the
quantum yield of fluorescence, representing the number of photons emitted by fluorescence per
absorbed photon. We note that photosystems I and II (PS1 and PSII, respectively) contribute to
leaf level fluorescence but only PSII is considered in models analyzed here (with the exception of
ORCHIDEE and BETHY, Section 2.3.4.2). $\phi_F$ is estimated as follows:
$$\phi_F = \frac{k_F}{k_F + k_D + k_N}(1 - \phi_P) \qquad\qquad \textit{Equation 2}$$
where *k* represents the rate coefficients for the different pathways for the transfer of energy
from excited chlorophyll ($k_F$ = fluorescence, $k_D$ = heat dissipation, and $k_N$ = non-photochemical
quenching, or NPQ), and $\phi_P$ is the quantum yield of electron transport (see Section 2.3.2). $k_F$ is
typically set to a constant value (0.05) in models following van der Tol et al (2014). $k_D$ is also
typically set to a constant value of 0.95, or temperature corrected in some cases (e.g., ORCHIDEE,
CLM4.5, CLM5.0, BETHY). $k_N$ has a substantial and variable impact on energy partitioning at
diurnal and seasonal scales which varies as a function of light saturation (e.g., Raczka et al., 2019;
Porcar-Castell et al., 2011). Once leaf level emissions are known, an approach is needed estimate
the total TOC fluorescence flux ($SIF_{canopy}$) for comparison to Photospec data. Leaf and canopy
level fluorescence modeling is described in more detail in Section 2.3.3.1 and 2.3.3.2 below.
*2.3.3.1 Leaf level SIF emission*
The 'quantum yield' approach has been used in SIF models to characterize the fraction of photons
that are used for PQ, NPQ, or re-emitted as fluorescence (van der Tol 2014). It is important to
note, that this does not translate into the actual amount of SIF emission leaving the leaf, but is
used as an approximation. TBM-SIF models typically represent $\phi_P$ using lake model formalism,
which assumes large connectivity between photosynthetic units (Genty et al., 1989; van der Tol
et al., 2014). $\phi_P$ is expressed in terms of the degree of light saturation (*x*), derived from the native
photosynthesis module of the parent TBM and represents the balance between actual and
potential electron transport rates, and the maximum photochemical yield under dark-acclimated
conditions ($\phi_{Pmax}$), which is derived from the fluorescence model and defined in terms of rate
coefficients in Eq 2.
$\phi_N$ accounts for the ability of plants to dissipate excess energy as heat via NPQ through the
regulation of xanthophyll cycle pigments (Demmig-Adams and Adams, 2006). NPQ can be
represented as a sum of reversible ($k_R$) and sustain ($k_S$) components ($k_N = k_R + k_S$). $k_R$ accounts for
the relatively fast (diurnal), reversible NPQ response to light. $k_S$ accounts for the relatively slow
(seasonal), sustained NPQ response to light and other environmental factors. With the exception
of CLM4.5, models do not typically account for $k_S$.
A significant challenge in fluorescence models is to find an appropriate relationship between $k_N$
and the degree of light saturation (x). The TBM-SIF models represent $k_N$ through an approach
similar to the one used in SCOPE, which uses a parametric model of $k_N$ derived from PAM
fluorometry measurements (van der Tol et al., 2014).
NPQ models can be classified as stressed (drought) and unstressed relative to water availability
depending on the dataset from which empirical fits are derived. The unstressed model is ideal
for irrigated systems such as crops, and the stressed model is more appropriate for water limited
ecosystems such as Niwot Ridge. We examine each of these models using drought and unstressed
models from van der Tol (2014), and a drought-based model from Flexas et al. (2002). These
models use different empirical fits but are otherwise identical. In general, $k_N$ increases more
rapidly with APAR (light saturation), and ramps up to a higher level, in the drought-based model
compared to the unstressed model. Additionally, some models provide unique improvements
such as dependence on environmental conditions (e.g., water stress vs no water stress in
ORCHIDEE), and equations for reversible and sustained NPQ to represent the different time
scales (minutes to seasonal) at which NPQ regulation occurs (e.g., CLM4.5) influenced by
pigmentation changes in the leaf.
*2.3.3.2 Leaf-to-Canopy scaling*
The TBM-SIFs produce leaf-level fluorescence which needs to be converted to canopy-level
fluorescence (SIF$_{canopy}$) to be directly compared to PhotoSpec and satellite observations. Leaf- to
canopy- level conversion of SIF requires a representation of canopy radiative transfer, which in
general is too computationally expensive to include within the TBMs in this study, that are
designed for global scale application.  Therefore, most TBMs analyzed here account for canopy
radiative transfer of SIF using some representation of SCOPE (van der tol 2009a,b). The most
commonly used approach is to run independent simulations of SIF from SCOPE to create an
empirical conversion factor ($\kappa_{740}$) between leaf and canopy level SIF that is a function of $V_{cmax}$
(Lee et al., 2015). This conversion factor accounts for integration over the fluorescence emission
spectrum, observation angle, and unit conversion. Model variations of this empirical approach,
as well additional approaches utilizing the full SCOPE model and a SCOPE emulator, are
summarized below and in Table 1.
*2.3.4 TBM-SIF Models*
Here we provide a brief description of individual TBM-SIF models and within model experiments.
We point out key differences in modeling of photosynthesis, fluorescence, and leaf-to-canopy
scaling. We note that within model experiments, labeled as Experiment 1 (exp1), Experiment 2
(exp2), etc, represent increasing order of realism, rather than a specific set of conditions common
across models. As such, Experiment 1 in BETHY (*BETHY-exp1*) is not equivalent to Experiment 1
in CLM4.5 (*CLM4.5-exp1*).
*2.3.4.1 BETHY*
The Biosphere Energy Transfer HydrologY (BETHY) model is the land surface component of the
Carbon Cycle Data Assimilation System (CCDAS) developed to ingest a range of observational data
for estimating terrestrial carbon fluxes at global scale (Rayner et al., 2005; Kaminski et al., 2013;
Koffi et al., 2012; Anav et al., 2015). Koffi et al. (2015) was the first to combine a process-based
model of SIF with a global TBM. The native canopy radiative transfer and photosynthesis schemes
of BETHY were effectively replaced with corresponding schemes and fluorescence model from
SCOPE (Koffi et al., 2015), thus enabling spatially explicit simulation of GPP and SIF as a function
of plant function type. This model was extended to include a module for prognostic leaf growth
(Norton et al., 2018) and more recently adapted with a formal optimization algorithm for
assimilating spaceborne SIF data (Norton et al., 2019). It has been updated for this study to accept
hourly meteorological forcing. BETHY-SCOPE, denoted here as BETHY, remains the first and only
global TBM-SIF model to simulate vertically integrated (1-D) fluorescence radiative transfer and
energy balance.
We include three experiments to examine the impact of calibrating the $k_N$ model against PAM
fluorometry data to different species: (1) *BETHY-exp1* is adapted to unstressed cotton species
(van der Tol et al., 2014), (2) *BETHY-exp2* is adapted to drought stressed Mediterranean species
(i.e., vineyard in controlled environment subjected to drought) including higher temperature
correction (Flexas et al., 2002; van der Tol et al., 2014), (3) *BETHY-exp3* is adapted to drought
stressed Mediterranean species (Flexas et al., 2002).
We further leverage SCOPE enabled SIF modeling in BETHY (*BETHY-exp3* specifically) to examine
(a) leaf and canopy level SIF and quenching under sunlit and shaded leaves, and (b) SIF emissions
at the top of canopy (SIF$_{canopy}$) versus the average emission within the canopy (SIF$_{ave}$), which
accounts for the average emission from sunlit and shaded leaves. The latter analysis facilitates
comparison to PhotoSpec, which observes the entire canopy.
An important caveat in the analysis of BETHY simulations is that, at the time of this writing, the
prescribed meteorological forcing at NR1 is only available for 2015. While this degrades
comparison to diurnal and synoptic variation observed by PhotoSpec in 2017, we find that
analysis of magnitude, light sensitivities, and within model experiments still provides useful
insight for interpretation of other TBM-SIFs, and future modeling requirements in general.
*2.3.4.2 ORCHIDEE*
The Organizing Carbon and Hydrology In Dynamic Ecosystems (ORCHIDEE) model (Krinner et al.,
2005) is the land surface component of the Earth System Model of Institut Pierre-Simon Laplace
IPSL-CM, (Dufresne et al., 2013) involved in recent exercises of the Coupled Model
Intercomparison Project (CMIP) established by the World Climate Research Programme
(https://www.wcrp-climate.org/wgcm-cmip). Recently a mechanistic SIF observation operator
was developed for ORCHIDEE to simulate the regulation of photosystem II $\phi_F$ at the leaf level
using a novel parameterization of NPQ as a function of temperature, PAR, and normalized $\phi_P$. It
emulates the radiative transfer of SIF to the top of the canopy using a parametric simplification
of SCOPE. The details of the SIF modelling approach are provided in Bacour et al. (2019).
We include three experiments to examine the impact of water stress and parameter optimization
(using OCO-2 SIF, see Section 2.4): (1) *ORCHIDEE-exp1* is the standard configuration with default
parameters, (2) *ORCHIDEE-exp2* is the same as *ORCHIDEE-exp1* with two key differences (a) water
stress is applied to stomatal conductance, mesophyll conductance and to the photosynthetic
capacity, and (b) the tree height (12 m instead of 15 m) was set specifically for the NR1 site, (3)
*ORCHIDEE-exp3* is the same as *ORCHIDEE-exp1* but includes OCO-2 optimized parameters.
*2.3.4.3 BEPS*
The Boreal Ecosystem Product Simulator (BEPS) is an enzyme kinetic two-leaf model for
simulating carbon and water cycles for different plant functional types (Chen et al., 1999; Liu et
al., 2003). BEPS uses a modified Ball-Berry stomatal conductance model (Leuning et al., 1995)
and semi-analytical canopy radiative transfer. The canopy architecture is well considered in BEPS
model, which has not only remote-sensed LAI but also the global map of the foliage clumping
index. The fluorescence emission at the leaf level follows the approach of Lee et al (2015). SIF
emission for sunlit and shaded leaves are separately simulated based on illumination and canopy
geometry in BEPS. In addition, multiple scattering SIF is also simulated to account for the
scattering process within the canopy. The scaling of leaf-level fluorescence emission to the
canopy is based on a novel scheme for single-layer models which accounts for canopy scattering
and extinction from sunlit and shaded leaves (Qiu et al., 2019). This scaling scheme is an effective
approach to simulate the radiative transfer of SIF for a given canopy structure. We include two
experiments similar to *BETHY-exp1/2* in the calibration of the $k_N$ model against unstressed vs
stressed species (*BEPS-exp1* and *BEPS-exp2*, respectively).
*2.3.4.4 CLM4.5*
The Community Land Model version 4.5 (CLM4.5) provides a description of the biogeochemical
profile spanning from the sub-surface bedrock to the top of the vegetation canopy. The
fluorescence sub-model follows Raczka et al. (2019), in which the degree of light saturation is
calculated from the potential and actual electron transport rate as determined from the
photosynthesis model described above. $\phi_f$ is formulated as described in Equation 2 and $\phi_P$ is
formulated as a function of the maximum $\phi_P$ under dark acclimated conditions and the degree
of light saturation. CLM4.5 uses independent site-level SCOPE simulations that match the
observed canopy characteristics and observed GPP at Niwot Ridge to calculate a leaf to canopy
level conversion factor ($\kappa_{740}$) for estimating $SIF_{canopy}$. In CLM4.5, $\kappa_{740}$ is fitted to the modeled
SCOPE data as a function of solar zenith angle (and implicitly $V_{cmax}$).
Similar to Raczka et al. (2019), here we examine three separate approaches to parameterize $k_N$.
*CLM4.5-exp1* only considers reversible NPQ ($k_R$), such that, $k_N = k_R$, and the relationship
between $k_R$ and the degree of light saturation is fitted to PAM fluorometry data based on
Mediterranean shrubs (Flexas et al., 2002; Galmes et al., 2007). *CLM4.5-exp2* parameterizes $k_R$
with PAM fluorometry from a Scots Pine forest (Porcar-Castell et al., 2011), and defines the rate
coefficient in terms of both a reversible and sustained component ($k_N = k_R + k_S$). It has been
found that sustained NPQ is important for cold climate evergreen conifer forests such as Niwot
Ridge (Miguez et al., 2015; Magney et al., 2019b), and Raczka et al. (2019) found that
representing both components provided improved simulations of seasonal SIF. *CLM4.5-exp3* is
similar to CLM4.5-exp3 but includes a seasonally varying representation of $k_R$. All model
experiments use hand-tuned parameters specific to US-NR1 (Raczka et al., 2016).
*2.3.4.5 CLM5.0*
CLM version 5.0 (CLM5.0) is similar to CLM4.5 with respect to the implementation of the
fluorescence sub-model, yet includes several important updates to the representation of
photosynthesis from CLM4.5, including a prognostic calculation of $V_{cmax}$ based upon leaf nitrogen
and environmental conditions, revised nitrogen limitation scheme, Medlyn stomatal
conductance model, and plant hydraulic water stress (Kennedy et al., 2019). To represent NPQ
we use a single approach for $k_N$ (see *CLM4.5-exp1*), but examine three approaches for estimating
$\kappa_{740}$: (1) *CLM5.0-exp1* uses $\kappa_{740}$ as function of $V_{cmax}$ following Lee et al (2015), (2) *CLM5.0-exp2*
follows the approach of *CLM4.5*, and (3) *CLM5.0-exp3* adapts the approach proposed by Zeng et
al. (2019) that estimates the fraction of total emitted SIF escaping the canopy by combining near-
infrared reflectance of vegetation ($NIR_V$) and fPAR.
*2.3.4.6 SIB3*
The Simple Biosphere Model version 3 (SIB3) involves the use of explicit biophysical mechanisms
to directly calculate carbon assimilation by photosynthesis (Baker et al., 2003; 2008). SiB3
includes prognostic calculation of temperature, moisture, and trace gases in the canopy air space,
but requires prescription of most structural properties including LAI. We examine two
approaches for prescribing LAI: (1) *SIB3-exp1* using values prescribed from MODIS, and (2) *SIB3-*
*exp2* uses values observed at the study site (4.0 $m^2$ $m^{-2}$). In general, the fluorescence sub-model
follows the approach of Lee et al. (2015) except that $k_N$ is adapted to drought stressed species
following van der Tol et al (2014).
*2.3.4.7 SIB4*
SIB4 (Haynes et al., 2019a,b) shares many similarities to SIB3 with respect to functional aspects
of photosynthesis and fluorescence, however, SIB4 uses prognostic rather than prescribed
phenology and LAI.
*2.3.5    SCOPE*
SCOPE is a multi-layer canopy model which explicitly represents the within canopy radiative
transfer of fluorescence, whereas TBM-SIFs analyzed here (with the exception of BETHY) only
provide an empirical representation. We provide results from a stand-alone version of SCOPE
v1.73 (van der Tol et al., 2014) as an additional benchmark for TBM-SIF simulations of APAR, GPP,
SIF, and quantum yields. There are three important reasons for this: (1) It is inherently difficult
to provide representative and accurate *in situ* measurements of APAR, SIF, and GPP for
comparison to models; (2) SCOPE provides estimates of quantum yields for fluorescence,
photochemistry, and non-photochemical quenching, which are not measured continuously in the
canopy at NR1; and (3) SCOPE offers a more direct benchmark for evaluating more simplified
representations of canopy radiative transfer in TBM-SIFs. Unlike the TBM-SIFs, SCOPE does not
include a representation of biogeochemical cycling or carbon pools, and thus no spin up is
required. As such, we prescribe LAI (4 $m^2$ $m^{-2}$), canopy height (13 m), and leaf chlorophyll content
(25 ug $cm^{-2}$) following Raczka et al. (2019). We also examine two approaches for prescribing $V_{cmax}$:
(1) *SCOPE-exp1* uses the default constant value of 30, similar to *BETHY*, and (2) *SCOPE-exp2* uses
a seasonal varying value calibrated to NR1, following Raczka et al. (2016, 2019), which follows a
bimodal distribution peaking near 45 in early summer (DOY = 150) and 40 in late summer (DOY =

499   250)

*2.4 Data Assimilation*

Details of the data assimilation protocols for ORCHIDEE is provided in Bacour et al. (2019). An
ensemble of parameters related to photosynthesis (including optimal $V_{cmax}$) and phenology were
optimized for several plant functional types. Note that none of the assimilated pixels encompass
the location of the US-NR1 tower. In ORCHIDEE, the study site is treated as boreal needleleaf
evergreen (ENF); as such, the *ORCHIDEE-exp3* simulations in this study are based on parameters
optimized against OCO-2 SIF data using an ensemble of worldwide ENF pixels. Note that for
BETHY, each experiment uses the same set of optimized parameters whereas in ORCHIDEE the
SIF simulations are performed separately for the standard parameters (*ORCHIDEE-exp1/exp2*)
and optimized parameters (*ORCHIDEE-exp3*), thus providing a test of sensitivity to parameter
optimization as discussed below.
*2.5 Illumination Conditions*
In order to gain insight into how SIF emissions and quantum yields vary with illumination, we
further analyze Photospec and a subset of models with respect to (a) changes in incoming light
and (b) self-shading within the canopy, respectively.  For PhotoSpec, we analyze changes in
canopy average SIF and $SIF_{rel}$ under conditions of predominantly direct versus diffuse PAR, using
a 0.5 threshold to distinguish between the two conditions (Section 2.2.3). For models we focus
on emissions from sunlit vs shaded leaves. We analyze leaf- versus canopy-level SIF emissions
($SIF_{leaf}$ and $SIF_{canopy}$) in *CLM4.5-exp3*, and leaf-level quantum yields ($\phi_f$, $\phi_p$, $\phi_N$) in *SCOPE-exp2*.
We further compare predictions of quantum yield at the top-of-canopy to canopy averages in
*SCOPE-exp2*. The motivation here is that top-of-canopy leaves see most of the sunlight, and thus
should have different yields compared to shade adapted leaves lower in the canopy. This also
provides a more direct comparison for PhotoSpec.
*2.6 Modeling Protocol*
Models are run for the period 2000-2018 (except BETHY (2015 only) and SCOPE (2017 only)) using
identical, hourly, gap-filled meteorological observations. The primary hourly output fields
analyzed are the top-of-canopy SIF ($SIF_{canopy}$ @ 740 nm), GPP, $\phi_f$, $\phi_p$, and APAR. Model-
observation comparisons are made for absolute and relative SIF, GPP, $SIF_{yield}$ ($SIF_{canopy}$/APAR) and
$GPP_{yield}$ (GPP/APAR), sunlit versus shaded canopies (*CLM4.5-exp3* and *SCOPE-exp2)*, and TOC
versus canopy average SIF ($SIF_{canopy}$ versus $SIF_{ave}$, respectively, from *SCOPE-exp2*). Quantum yields
and within model experiments provide context to understand canopy integrated results. We
focus our analysis on 8 am – 4 pm local time from July-August 2017 for comparison to available
PhotoSpec and APAR data.
Models are controlled for meteorological forcing (meteorological data described in Burns et al.,
2015) but other factors such as spin-up, land surface characteristics, parameter tuning, and
model state, are not controlled for and are treated separately according to each model's
protocol. For example, CLM4.5 is better suited than others in prescribing observed vegetation
characteristics at the study site. One ORCHIDEE experiment (*ORCHIDEE-exp3*) is preliminary
optimized by assimilating independent Orbiting Carbon Observatory 2 (OCO-2) SIF data at the
global scale (Section 2.4). We emphasize that our point here is not to identify the best model but
to identify common patterns in model behavior through normalized SIF and deviation from
observed behavior to identify areas requiring the most attention.
The results are organized around two parallel themes. The first theme addresses four key
processes driving canopy-level fluorescence: (1) incoming illumination, (2) energy partitioning on
incoming light between photochemistry, fluorescence, and NPQ, and (3) leaf-to-canopy emitted
SIF, including linearity of yields at leaf and canopy scale. The second theme addresses sensitivity
of these processes to environmental conditions at diurnal and synoptic scales. Here, synoptic
scale refers to the impact of day-to-day changes in weather, including two storm events which
brought sustained cool, wet, and cloudy conditions from July 22-31 and then from August 6-10.
**Section 3: Results**
*Incoming Illumination*
Two key features dominate observed APAR variability: afternoon depression (Fig 2A) and
reduction during two summer storms (Fig 2D). Both features are captured by models. More
generally, models capture synoptic variability with high correlation (r > 0.8) and low across model
spread ($\sigma$ = 10%). The exception is BETHY, which is simulated outside our observation year (2015).
High model fidelity is expected given that observed PAR is prescribed, and it is promising that
models show a consistent response to changes in illumination. The primary shortcoming across
TBM-SIFs and SCOPE is a systematic high bias in APAR magnitude (129%), with most models
exceeding the upper range of observed APAR (as determined from the six within canopy PAR
sensors, Fig S2), and high model spread. These errors are likely related to differences in predicted
fAPAR. In the case of ORCHIDEE, high APAR is expected due to the big leaf assumption where all
leaves are considered as opaque and fully absorbing.
*Canopy Photosynthesis*
Observed GPP shows a broad peak from mid-morning to early afternoon (~9 am – 1 pm local),
followed by slight decrease until 4 pm (Fig 2B), consistent with afternoon cooling and reduced
light availability (Fig 1B-D). The two month period under investigation is relatively flat with
generally weak day-to-day variability ($\sigma$ = 17%), but modest correlation with APAR (r = 0.61, Fig
2E). Some models capture the afternoon GPP depression, but all models strongly underestimate
its magnitude, apparently independent of stomatal conductance formulation or more explicit
accounting for plant hydraulic water stress such as in CLM5.0. *SCOPE* and *BETHY*, which don't
account for water stress, show no afternoon depression. Models are mostly uncorrelated with
observed GPP at synoptic scale (*r* ranges from -0.2 to 0.36, highest value in SiB4), high biased,
and show increased spread (in predicted magnitude) relative to APAR (143% +/- 23%). *SCOPE-*
*exp2* shows slight improvement in GPP magnitude with the larger $V_{cmax}$ value in late summer.
While observed GPP$_{yield}$ is mostly stable over the diurnal cycle, most models (except BEPS) show
a distinct midday minimum (Fig 3A). Half of the models show a similar midday minimum in
photochemical quantum yield ($\phi_P$, Fig 4A), with the other half either increasing or decreasing in
the afternoon (CLM5.0 and SiB3/SiB4, respectively). The midday dip in yield is likely associated
with reduced photosynthetic efficiency at high light levels, as demonstrated by reductions in GPP,
GPP$_{yield}$, $\phi_P$ with APAR (Fig 5A, C, E).
Observed GPP$_{yield}$ shows significant structure at synoptic temporal scale (Fig 3C), most notably
increased yield during the cool/rainy period (reduced heat and water stress), and decreased yield
in mid- to late- August (increased heat and water stress following the cooling pattern). In contrast
to predicted GPP, models show high fidelity in capturing the magnitude and variability of $GPP_{yield}$
at synoptic scale ($r$ ranges from 0.35 – 0.76, highest values in *SCOPE* and *CLM4.5/5.0*). Individual
models are self-consistent in their predictions of $GPP_{yield}$ and $\phi_P$ at synoptic scale (r = 0.592 –
0.935) except for SiB3/SiB4 (r < 0.1, Fig 4B).

*Canopy Fluorescence*

Observed $SIF_{canopy}$ is strongly correlated with observed APAR at diurnal and synoptic scale (r =
0.77), with common features including afternoon depression and reduction during rainy periods
(Fig 2C & 2F). Observed PAR also feeds into the fluorescence sub-model and, unlike GPP, strongly
correlates with $SIF_{canopy}$ at synoptic scale (*r* ranges from 0.58 to 0.92, highest values in *SCOPE and*
*ORCHIDEE*). However, we find a persistent positive model bias in $SIF_{canopy}$ (170% +/- 45%)
consistent with, but not proportional in magnitude to, the APAR bias. We note that models are
especially oversensitive to APAR at high light levels (Fig 5D).
We investigate the high bias in $SIF_{canopy}$ in more detail using *SCOPE-exp2* and *CLM4.5-exp3*.
Specifically, we examine leaf and canopy level SIF and quenching under sunlit and shaded leaves.
Analysis of quantum yields in *SCOPE-exp2* (Fig S5) shows a reversal in the fractional amounts of
absorbed energy going to SIF and PQ vs NPQ in low- vs high-light conditions that is consistent
with leaf level data and theory (Porcar-Castell et al., 2014). More specifically, *SCOPE-exp2*
predicts low $\phi_F$ and $\phi_P$ and high $\phi_N$ in sunlit leaves relative to shaded leaves, with more energy
going to fluorescence and photochemistry than to NPQ in shaded leaves, and more energy going
to (shed off by) NPQ in sunlit leaves (Fig S5). Likewise, total $\phi_F$ shows decreasing values with
increasing APAR in *SCOPE* and *BETHY-exp2/3* compared to *BETHY-exp1*, consistent with observed
$SIF_{yield}$ (Fig 5E-F), as $\phi_N$ ramps up to higher levels in the drought parameterized Kn model.
Moreover, in stark contrast to $SIF_{yield}$ and $SIF_{canopy}$, $\phi_F$ does not show high values relative to other
models (Fig 4D). These results point to an issue in *SCOPE* and *BETHY* with leaf to canopy scaling
in needleleaf forests.
Analysis of *CLM4.5-exp3* suggests several possible reasons for oversensitivity to APAR. First, we
focus on emissions from sunlit/shaded portions of the canopy (Fig S6). *CLM4.5-exp3* and
PhotoSpec both show higher SIF under "high light" conditions (sunlit leaves and direct radiation,
respectively) compared to "low light" conditions (shaded leaves and diffuse radiation,
respectively), which is promising (Fig S6 A,D). Comparing the ratio of sunlit to shaded SIF in
*CLM4.5-exp3* to the ratio of direct to diffuse SIF in PhotoSpec (Fig S6 B,E) shows higher ratio in
*CLM4.5-exp3* on average. The difference peaks in midday, when sunlit leaf area is maximized
(self-shading minimized) in CLM4.5 but no major difference in the amount of direct radiation,
and decreases with increasing sun angle (morning and afternoon) and with increasing rainfall (in
the afternoon on average, and during the rainy period in late July / early August), both of which
increase the shaded fraction. As such, accounting for view angle and different illumination
metrics for PhotoSpec and CLM4.5 (most comparable in morning, afternoon, and during rainy
days) reduces, but does not entirely remove, the positive bias in high light conditions.
Second, the degree of light saturation ($x$) is twice as high in the sunlit canopy in *CLM4.5* (Fig S7),
which leads to low fluorescence efficiency in sunlit leaves and high fluorescence efficiency in
shaded leaves. While this produces high photochemistry in shaded leaves, it contributes a small
fraction of SIF to the total canopy (~20%) despite higher fractions of shaded leaves (~2/3 at noon,
Fig S6C) and thus sunlit leaves dominate $SIF_{yield}$ and $SIF_{canopy}$.  Therefore, it seems likely that a
model's representation of canopy structure including the partitioning between sunlit/shaded leaf
area fractions has an important impact upon canopy SIF.  Biases in the sunlit/shaded fraction will
likely propagate into the simulated value of canopy SIF. However, it's important to know that the
observed sunlit/shaded fraction from PhotoSpec is estimated as well, since it is currently not
possible to determine the precise sun/shade fraction within PhotoSpec FOV.
Additionally, all formulations of CLM4.5 (and most models except BETHY and SCOPE) show lack
of decline in $SIF_{yield}$ with APAR compared to measurements of absolute SIF (Fig 5E). For CLM4.5,
the relationship between $SIF_{yield}$ and APAR depends upon the relationship between degree of
light saturation and reversible NPQ (Raczka et al., 2019).  This suggests it is important to properly
represent the NPQ response to environmental conditions when simulating SIF.
While most of the model bias is reduced in $SIF_{yield}$ (126%, mostly attributed to BETHY and SCOPE),
the remaining signal, representing the dynamic response to synoptic conditions (e.g., Magney et
al., 2019), is poorly represented in models, as demonstrated in a time series of 5-day means (Fig
3D). Most models show zero to strongly negative correlation with observations at synoptic scale
and only three models (*SCOPE, ORCHIDEE-exp3,* and *BETHY-exp2/3)*, produce correlation greater
than 0.5. These are the only three models that also capture a negative relationship between
$SIF_{yield}$ and APAR (Fig 5E).
In general, predicted $SIF_{yield}$ is stable during our short study period (Fig 3). Half of models show a
significant positive correlation with $GPP_{yield}$ (r > 0.85) and half show zero or negative correlation
(Fig S8). While these findings run counter to observed $SIF_{yield}$, which shows a clear response
during and following the storm event and moderate positive correlation with observed $GPP_{yield}$ (r
= 0.40), they show some consistency with observed $SIF_{rel}$ (grey line in Fig 3 and Fig S8A) which
like many models is stable and uncorrelated with $GPP_{yield}$. We refer the reader to Section 2.2.2
for clarification of the important difference between $SIF_{yield}$ and $SIF_{rel}$.
*Leaf-to-Canopy Scaling*
Several methods have been proposed to transfer predicted leaf-level SIF emissions to the top of
canopy. While leaf-to-canopy scaling enables efficient global scale simulation, the diversity of
novel methods adds uncertainty to the canopy level estimate of SIF (in addition to
aforementioned uncertainties in structure, APAR, photochemistry, fluorescence). These
differences are evident in comparison of Figures 3 and 4, in which yields are plotted on a similar
scale.
At least at diurnal scale, there is some evidence that leaf and canopy emissions look more similar
for models adopting simplified empirical scaling functions (SiB3, SiB4, CLM4.5, CLM5.0, BEPS)
than for models that more explicitly account for radiative transfer (SCOPE, BETHY, ORCHIDEE).
For the more explicit models, the diurnal cycle of $\phi_f$ is out of phase with $SIF_{yield}$, the former of
which peaks in the afternoon and the latter of which peaks in the morning. This produces
reasonable agreement to PhotoSpec in phase and magnitude between $SIF_{yield}$ and $SIF_{rel}$ for
ORCHIDEE, but produces divergence in the magnitude of $SIF_{canopy}$ for ORCHIDEE.
Model performance in leaf-to-canopy scaling is summarized in Figure S8. The only three models
with a positive relationship between yields (Fig S8B) and between quenching terms (Fig S8C)
include explicit representation of radiative transfer (i.e., SCOPE, BETHY, and ORCHIDEE). CLM4.5
is the only model with a positive relationship between yields, but not between quenching terms.
SiB3/SiB4 are the only models with a positive relationship between quenching terms, but not
between yields.
Finally, we clarify an important difference between observed and predicted estimates of canopy
average SIF. PhotoSpec scans direct emissions from sunlit and shaded leaves within the canopy,
thus observing the 'total' emission from leaves in the instrument FOV. We then average each of
these leaf-level scans and report as canopy averages. Model output, in contrast, is reported at
the TOC, which represents the 'net' emission from leaves after attenuation in the canopy
(through canopy radiative transfer, re-absorption of SIF, and shading). Assuming sunlit and
shaded leaves within the canopy emit at the same rate as TOC leaves, attenuation will reduce the
effective signal from leaf-level emissions within the canopy. As such, the average of leaf level
emissions (canopy average) is expected to be lower than the net emission of leaves reaching the
top of canopy.
This is important because CLM4.5 shows strong attenuation of SIF from leaf-level to TOC,
decreasing by a factor of 2-3 at midday (Fig S7). The interpretation here is that the model bias in
absolute SIF may actually be higher than reported here; however, we note that more quantitative
information on the observed fraction of sunlit vs shaded leaves and comparative top-of-canopy
SIF values for the same canopy elements are needed (to account for off-nadir SIF viewing) for
more accurate determination of scaling between observed canopy and top-of-canopy SIF.
*Within Model Experiments*
In most cases, within model experiments produce improvements in some metrics and
degradation across others (performance change is quantified by reporting correlation values in
brackets). An important and unexpected result of this study is the impact of different levels of
tuning to observations on our predictions. While this work represents a snapshot of the state-of-
the-art in site-level TBM-SIF modeling, and we have taken great care to control for environmental
conditions (most important being illumination), an important overall takeaway is for future
model comparisons to make additional efforts to control for initial conditions and vegetation
state (i.e. model biophysical parameters).
The most basic example is tuning of LAI in SiB3 and $V_{cmax}$ in SCOPE. LAI, as prescribed by MODIS
for *SiB3-exp1* (~1.5), is on the low end for a subalpine evergreen forest, and consequently
produces negative biases in APAR, GPP, SIF and $SIF_{yield}$. When prescribed according to tower
observations in *SiB3-exp2* (~4.0), the biases become positive (albeit on the lower end of the
model ensemble), but produces degraded variation at synoptic scale for GPP (0.39 vs 0.19), SIF
(0.87 vs .71) and $SIF_{yield}$ (0.09 vs -0.32). The tuning of $V_{cmax}$ in SCOPE improves the magnitude of
GPP, with minimal impact on variability at diurnal- to synoptic- scale.
Experiments in CLM4.5 comprise a higher level of hand tuning of vegetation structural and
functional characteristics.  Parameter tuning was imposed to match vegetation structure with
site level measurements and consequently CLM4.5 produces overall low bias in yields. With
respect to synoptic variation, NPQ experiments, tuned against the measured air temperature and
a representative evergreen forest, produce improvements at synoptic scale for GPP (-0.01 vs
0.16), SIF (0.59 vs 0.86), and $GPP_{yield}$ (0.05 vs 0.63), but degradation in $SIF_{yield}$ (0.32 vs -0.25).
Likewise, NPQ experiments in BETHY based on species information (calibration of $K_N$ against PAM
fluorescence in stressed vs unstressed systems) shows improvement in the $SIF_{yield}$-APAR
relationship for drought stressed models (*BETHY-exp1* vs *BETHY-exp2/3*).
Experiments with ORCHIDEE demonstrate that errors in model parameters (such as $V_{cmax}$, $LAI_{max}$,
leaf age, or SLA) contribute to SIF and GPP uncertainty but can be alleviated by assimilation of
OCO-2 SIF retrievals (*ORCH-exp1/2* vs *ORCH-exp3*). Model optimization of parameters improves
the functional link between SIF and GPP, thus reducing biases in APAR, GPP, and $SIF_{yield}$, and
improving synoptic variation in $SIF_{yield}$ (-0.04 vs 0.58).
**Section 4. Discussion**
This study represents a first attempt to evaluate a controlled ensemble of TBM-SIF models
against canopy integrated SIF observations to identify and attribute model-observation
mismatches related to errors in canopy absorption of sunlight, photosynthesis, fluorescence, and
leaf-to-canopy radiative transfer of fluorescence.
Different models match some observed parameters better than others (with respect to APAR and
yield), but no model gets both APAR and $SIF_{yield}$ magnitude and/or sensitivities close to the
observations. For example, BEPS closely matches the magnitude of APAR (Fig 2A), and BETHY
captures the decline in SIF$_{yield}$ with APAR for NPQ quenching based on stressed species (Fig 5E),
but both models overestimate observed yield by a factor of 2, hence SIF is overestimated (Fig 2).
CLM4.5 correctly captures the diurnal SIF$_{yield}$ change, but overestimate APAR; in this case, SIF and
SIF$_{yield}$ are overestimated. Importantly, models diverge strongly from each other and from
observations in the magnitude of SIF$_{yield}$ and its decline with APAR (Fig 5E), partially reflecting
model variability in $\phi_f$ (Fig 5F), but in general show a characteristic pattern of weak SIF$_{yield}$ decline
with APAR. GPP$_{yield}$ shows higher agreement between models and with observations (Fig 5B),
despite divergent $\phi_P$ (Fig 5C), which could be indication that the primary uncertainty is due to
the representation of fluorescence and not the photosynthesis model.
Consequently, we find a strong linear and positive relationship between observed SIF$_{yield}$ and
GPP$_{yield}$ for absolute SIF, which is underestimated on average by models (Fig S8A-B). In contrast,
models show quite strong positive relationships between $\phi_f$ and $\phi_P$ (Fig S8C). Our study
highlights an apparent challenge for models in transferring leaf level processes to canopy scale,
and consequently, linking the proper canopy mechanistic SIF-GPP relationship at the leaf level.
The mismatch between multi-model simulations and tower-based observations of SIF and GPP
at hourly and daily scales can be summarized as symptoms of five main factors: (1) PhotoSpec
scan strategy, (2) radiative transfer of incoming PAR and impact on APAR and sunlit/shaded
fraction, (3) representation of photosynthesis and sensitivity to water limitation especially during
afternoon conditions, (4) representation of fluorescence and sensitivity to reversible NPQ
response at Niwot Ridge, and (5) radiative transfer of fluorescence from leaf to canopy. Several
persistent biases falling under these broad categories are discussed below.
**Apples to Apples Comparison**.
PhotoSpec is unique in its ability to scan entire canopies for signals that are largely hidden from
nadir-oriented instruments. However, this creates unique challenges for interpretation of data
and comparison to models. For example, the diurnal cycle of observed SIF is highly sensitive to
view angle. PhotoSpec was set up in 2017 to scan back-and-forth between northwest and
northeast view angles, but the instrument was slightly biased to the northwest, causing a low
phase angle in the morning (more aligned with rising sun) and increased phase angle in the
afternoon (more opposed to setting sun). As such, PhotoSpec observed predominantly
illuminated canopies in the morning and shaded canopies in the afternoon (i.e., more shaded
fraction), leading to the late morning peak in reflected radiance (Fig S3).
Moreover, Photospec scans specific locations at the top of the canopy from near nadir to view
angles closer to the horizon (see Fig. S8 in Magney et al., 2019b), while models are currently
configured to simulate top of canopy emission and simulated here as nadir viewing. The question
becomes whether to retain nadir only data and sacrifice signal-to-noise, or to average over all
elevation angles and risk aliasing view angle effects. This study, partly motivated by high
agreement of canopy integrated SIF with spaceborne data from OCO-2 and TROPOMI (Magney
et al., 2019b; Parazoo et al., 2019), has chosen the latter approach but with an attempt to
minimize scan angle effects in $SIF_{rel}$. However, it is worth noting that swath sensors such as
GOME-2 show high sensitivity to viewing angle especially under increasing illumination angles
(Kohler et al., 2018; Joiner et al., in review). View angle effects are likely to be especially acute
for PhotoSpec in the morning and afternoon with increasing anisotropy and changes in the
illuminated field of view with sun and view angle. Other tower SIF instruments with a wide FOV
(i.e. FluoSpec2; Yang et al., 2018) may more appropriately represent the TOC SIF emission, but
also have difficulty disentangling the sunlit/shaded canopy components.
It is critical that model evaluation relative to measured SIF data and data assimilation studies
properly account for the specificities of the instrument (viewing of the instrument, spectral band,
time of the overpass for space-borne instruments), the representation of canopy emission, and
correct observations for directional variations in SIF relative to observation geometry. Although
normalizing SIF by reflected radiance partially alleviates scan angle effects, this highlights the
need for models to get canopy structure, radiative transfer, and sunlit/shaded fraction correct,
which feed all the way through to SIF and GPP. Further ground-based investigations of SIF
anisotropy, sunlit/shade fraction, and vertical distribution (within canopy, canopy integrated,
and top of canopy) with PhotoSpec and SCOPE may help to inform models on the physical aspects
of the signal. Despite the issues we highlight in comparing observations to models, the potentially
more interesting and important story here is with respect to model-model comparisons, which
reveals wide divergence in response to light conditions and other factors, as discussed below.
**TBM SIF is too sensitive to APAR.**
Our results indicate a wide range of SIF responses to APAR: TBM-SIFs and SCOPE are usually far
too sensitive to APAR, observations of absolute SIF are less sensitive, and observations of relative
SIF ($SIF_{rel}$) are least sensitive (Fig. 5D). We remind the reader that $SIF_{rel}$ is normalized by the
amount of far-red light reflected from leaves in the FOV of PhotoSpec, and thus has reduced
sensitivity to absorbed light than absolute SIF. The fact that $SIF_{rel}$ is the least sensitive to APAR
means other processes are driving changes in SIF under increased light absorption. In this case,
it reveals a strong SIF response to changes in photochemical quenching. SIF models appear
especially sensitive to sunlit leaves. In CLM4.5, SIF emissions from the sunlit portion of the canopy
are a factor of 5 higher than emissions from shaded leaves, despite twice as fewer leaves in the
sunlit canopy (Fig S6C). In CLM4.5, the combination of higher than average $\phi_f$ (Fig 5F) with higher
fluorescence efficiency in the sunlit portion of the canopy, produce an increase in the magnitude
and sensitivity to sunlit fraction, thus contributing to the high bias (factor of 3 higher than
observed) and strong diurnal cycle (2-fold increase from morning to midday).
**Linearity of SIF and GPP yields**.
Observations show a positive but not significant linear relationship between $SIF_{yield}$ and $GPP_{yield}$
(Fig 8A, r = 0.40) at our study site. This is likely due to the short time period investigated here
where there is relatively little change in $SIF_{yield}$ and $GPP_{yield}$ during peak summer. Half of models
(4 of 8) show a significant (r > 0.35) linear and positive slope (r > 0.35; SCOPE, ORCH-exp3,
CLM4.5-exp3, and BETHY-exp3) between $SIF_{yield}$ and $GPP_{yield}$, while 6 models (except CLM5.0)
show a significant positive slope between quantum yields ($\phi_f$ and $\phi_p$, Fig S8C). These regression
plots of quantum yields, in turn, help explain the observed linearity of $SIF_{yield}$ vs. $GPP_{yield}$: At least
in the case of Niwot Ridge, model (and presumably observed) $\phi_p$ stays within high light "NPQ-
Phase" conditions, and generally doesn't exceed the range in which decoupling of $\phi_f$ and $\phi_p$ ($\phi_p$
> 0.6) in low light "PQ-Phase' conditions occurs (Porcar-Castell et al., 2014, cf Fig 9). SCOPE and
BETHY-exp3, which best capture the observed relationship in the canopy between $SIF_{yield}$ and
GPP$_{yield}$, are also the only models that also show a decline in SIF$_{yield}$ with APAR, as discussed below.
These results are likely to change when we expand the study to several years; however, the
purpose of this study was to provide an initial investigation into the response of modelled SIF and
GPP to light during peak summer.
**Insufficient decline in SIF$_{yield}$ with APAR.**
In general, models show an insufficient decline in SIF$_{yield}$ with APAR, when compared to observed
SIF$_{yield}$ (Fig 5E). All models except SiB3 and SiB4 show some decline, with BETHY showing the best
agreement in slope magnitude. SCOPE and BETHY are the only models with full radiative transfer
but this does not appear to have a substantial impact on SIF$_{yield}$, which has a similar (albeit
suppressed) decline with APAR as $\phi_f$ (Fig 5F). Within model experiments show little to no
sensitivity of SIF$_{yield}$ or $\phi_f$ decline with APAR to water stress (e.g., ORCHIDEE) or prescribed LAI
(e.g., SiB3), but high sensitivity to the formulation of NPQ with respect to species calibration (e.g.,
BETHY) and reversibility (e.g., CLM4.5).
Three CLM4.5 experiments demonstrate sensitivity to representation of NPQ variability at diurnal
and seasonal scales. The first simulation using the default NPQ parameterization from SCOPE
(*CLM4.5-exp1*, based on a 2-parameter fit to drought stressed Mediterranean species (Galmes et
al., 2007) produces the strongest decline in SIF$_{yield}$. The second simulation, which includes a site-
specific NPQ formulation that accounts for k$_R$ and k$_S$ (*CLM4.5-exp2*), produces the weakest
decline. The third simulation with seasonally varying k$_R$ produces a slightly stronger decline. An
important point for this formulation is that k$_R$ is constrained by PAM fluorometry data at Hyytiala
(Scot Pine) and does not account for high light saturation values and summer drought conditions
that may be more typical of lower latitude sites such as Niwot Ridge. This could indicate that
parameterizing k$_R$ based upon similar PFTs may not be sufficient to properly characterize the NPQ
response for lower latitude sites such as Niwot Ridge.
Similar results are found in experiments with BETHY comparing stressed (drought) and
unstressed (relative to water availability) NPQ models at NR1 but controlling for k$_R$ (constant in
time in both cases, stronger negative *SIF$_{yield}$* response to APAR in stressed model). In the
unstressed models of CLM4.5 and BETHY, the NPQ response to APAR becomes too low, causing
an oversensitivity of SIF to APAR and thus high SIF bias. The strongly regulated NPQ response of
the drought-based model enables more non-photochemical quenching at high light levels in
stressed ecosystems compared to typical unstressed plants. While this $k_{NPQ}$ model was
developed using drought-stressed plants, similar up-regulation of NPQ is expected to occur under
any condition where photosynthesis is limited and available excitation energy is high (e.g. cold
temperatures and high light, Sveshnikov et al., 2006). Our results thus emphasize the need for
careful implementation of NPQ dynamics for simulating and assimilating SIF in different light and
stress environments (Raczka et al., 2019; Norton et al., 2019).
**Data assimilation reduces high bias**. Assimilation of OCO-2 SIF in ORCHIDEE brings the magnitude
of both GPP and SIF in closer agreement with observations. This improvement is driven by
decreases in leaf photosynthetic capacity (V$_{cmax}$, LAI$_{max}$, leaf age, SLA, Bacour et al., 2019), which
decreases the magnitude (but not shape) of APAR closer to observed values (Fig 2), and leads to
improvements in GPP$_{yield}$ and SIF$_{yield}$ (Fig 3). Nevertheless, after the assimilation there are still
disagreements in SIF$_{yield}$ vs GPP$_{yield}$ relative to the measured quantities (Fig S8). For diurnal and
synoptic cycles, the assimilation effectively acts to scale the magnitude of SIF, GPP and APAR (and
related yields), but it does little to alter variability. Although data assimilation (i.e. calibrating
model parameters) is critical to improving modelled SIF and GPP, this should be done in
conjunction with improvements in the model formulation (as summarized in Section 5),
otherwise the estimated model parameters can be sub-optimal to compensate for the lack of
missing processes.
**5. Conclusions/Recommendations**
Our results reveal systematic biases across TBM-SIF models affecting leaf-to-canopy simulations
of APAR, GPP, and SIF. This highlights key areas where observing strategies and model
formulations can be improved:
1)  Radiative transfer of incoming and absorbed PAR. The representation of incoming radiative

transfer produces positive biases in APAR that leads to positive biases in GPP, both of which

occur regardless of time of day. This is influenced by characterization of the canopy, leaf

orientation and clumping, biochemical content, canopy layers, and leaf area, which dictates

the sunlit/shaded fractions of the canopy. Furthermore, the combination of high APAR bias
in models and high uncertainty in observed APAR highlights a need for more accurate and
representative *in situ* measurements of APAR within the FOV of SIF observations and
footprint of eddy covariance data. We recommend further site-level investigation of
observed and simulated canopy light absorption, emphasizing comparison of multi-layer and
multi-leaf radiation schemes accounting for sunlit and shaded leaf area.
2) Water stress impacts on photosynthesis. The underlying photosynthetic models fail to
simulate the magnitude of depression of observed GPP in the afternoon, regardless of how
stomatal-conductance and water stress models and parameters are formulated. This likely
results from the inability to account for afternoon water stress to properly restrict stomatal
conductance and hence GPP and SIF.  Additional effort is needed to characterize SIF and GPP
sensitivity to increased atmospheric demand and/or reduced soil moisture across a range of
managed and unmanaged systems. We also recommend more inclusion of stomatal
optimization models (e.g., Eller et al., 2020) as optional parameterizations for TBMs, to better
account for plant hydraulic functioning under water stress compared to the more widely used
semi-empirical models.
3) Leaf Mechanism for Energy Partitioning. We provide evidence that many models fail to
capture the correct reversible NPQ response to light saturation, leading to biases in $SIF_{yield}$
during high light conditions and especially with increasing moisture limitation at the end of
summer. Further investigation using models such as BETHY and CLM is needed to better
characterize sensitivity of NPQ formulations to PFT and environmental conditions. We also
emphasize a need for more simultaneous measurements of active and passive chlorophyll
fluorescence to determine the temporal dynamics of competing pathways (PQ, NPQ) from a
wider variety of plant species under ambient conditions and different levels of stress.
4) Radiative transfer of SIF. SIF is emitted from the leaf level and then is transferred to the top
of canopy as a function of canopy structure (leaf geometry, canopy layers, leaf area,
sunlit/shaded fraction). Despite high disagreement of SCOPE and BETHY with respect to the
simulation of APAR and SIF magnitude, we recommend site level simulations using a similar
framework where a radiative transfer model is run both offline and coupled to a terrestrial
biosphere model for more detailed investigation of sensitivity to canopy characteristics.
5) Observation strategy. The PhotoSpec scan strategy enables direct measurement of SIF
emission at leaf-to-canopy scale, but requires off-nadir view angles that lead to changing
fractions of sunlit and shaded canopies throughout the day as a function of sun angle. Further
work could be done using tower mounted instruments with a wider FOV that more accurately
represent top of canopy emissions for comparison to model simulations, and to classify
emissions from shaded vs sunlit canopies. More effort is also needed to better align models
with observations, for example by leveraging three-dimensional capabilities in SCOPE (and
other RTMs) to directly account for multiple observation angles.

6) Finally, we note that our focus on a water limited subalpine evergreen needleleaf forest
represents a challenging case study for models and observations. In many cases, there is
strong covariance between LAI, SIF, APAR and GPP in cropping systems (Dechant et al., 2020),
but because this study site experiences little change in canopy structure and APAR
throughout the season (Magney et al, 2019b), our study sought to provide more explicit
insight into the models sensitivity to photosynthesis and fluorescence. As such, it is possible
that we would see more convergence of results, and a reduction in confounding effects (e.g.,
decreased NPQ), in a well-watered high-LAI cropping system. We therefore recommend
similar model-observation assessments across a wider range of biota and climate.

**Data availability**

All observational data (APAR, SIF, GPP, and relative SIF) are provided as hourly time series. The
data can be found at https://data.caltech.edu/records/1231. The data are saved as a .csv file.

**Author Contribution**

NP, TM, and IB designed research. NP, TM, AN, BR, CB, FM, IB, YZ, BQ, MS, DB performed
research; AN, BR, CB, FM, IB, YZ, BQ, MS, NM contributed model simulations; TM, DB, SP, PB, JS,
KG, CF contributed observational data; NP, TM, AN, BR analyzed data; NP, TM, AN, BR, CB, IB, YZ,
NM, DB, CF wrote paper.

**Competing Interests**

The authors declare that they have no conflict of interest.

**Acknowledgements**

The US-NR1 AmeriFlux site is supported by the U.S. DOE, Office of Science through the AmeriFlux Management Project (AMP) at Lawrence Berkeley National Laboratory under Award Number 7094866. BMR was supported by the NASA CMS Project (award NNX16AP33G) and the US Department of Energy's Office of Science, Terrestrial Ecosystem Science Program (awards DE-SC0010624 and DE-SC0010625). CESM (CLM4.5 and CLM5.0) is sponsored by the National Science Foundation and the U.S. Department of Energy. ORCHIDEE is supported by CNES-TOSCA under the FluOR and ECOFLUO projects. ITB was supported by NASA contract 80NSSC18K1312. We would like to thank the W.M. Keck Institute for Space Studies and internal funds from the Jet Propulsion Laboratory for support of the field measurements at Niwot Ridge (http://www.kiss.caltech.edu/study/photosynthesis/technology.html). A portion of this research was carried out through the OCO-2 project at the Jet Propulsion Laboratory, California Institute of Technology, under contract with NASA. This work was supported in part by the NASA Earth Science Division MEaSUREs program (grant 17-MEASURES-0032) and ABoVE program (18-TE18-0062).

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

**Figures**

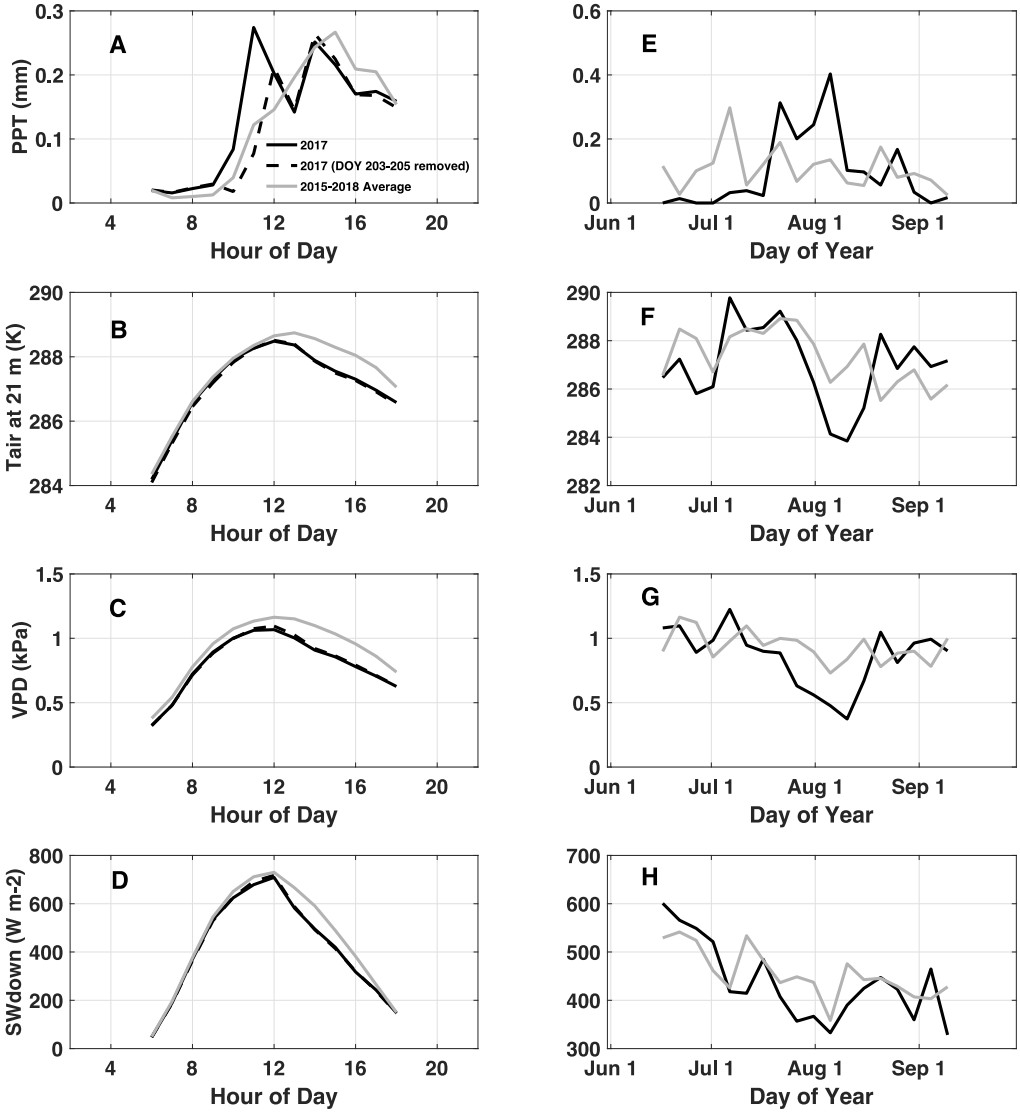


**Figure 1**. Observed diurnal (A-D) and synoptic (E-H) precipitation (PPT), air temperature at 21 m
(Tair), vapor pressure deficit (VPD), and downwelling shortwave (SWdown). Diurnal cycles are
averaged over July-August, 2017. Synoptic cycles are plotted as 5-day averages from June 15 –
Sep 15. Data from 2017 is shown in black and climatology (2015-2018) in grey. Typically, peak
rainfall occurs in the afternoon at this site (A ). A substantial rain event which occurred from DOY
203-205 is removed from the 2017 average to show the impact on diurnal variability and to
demonstrate the dominance of the afternoon monsoon upon diurnal precipitation in summer.

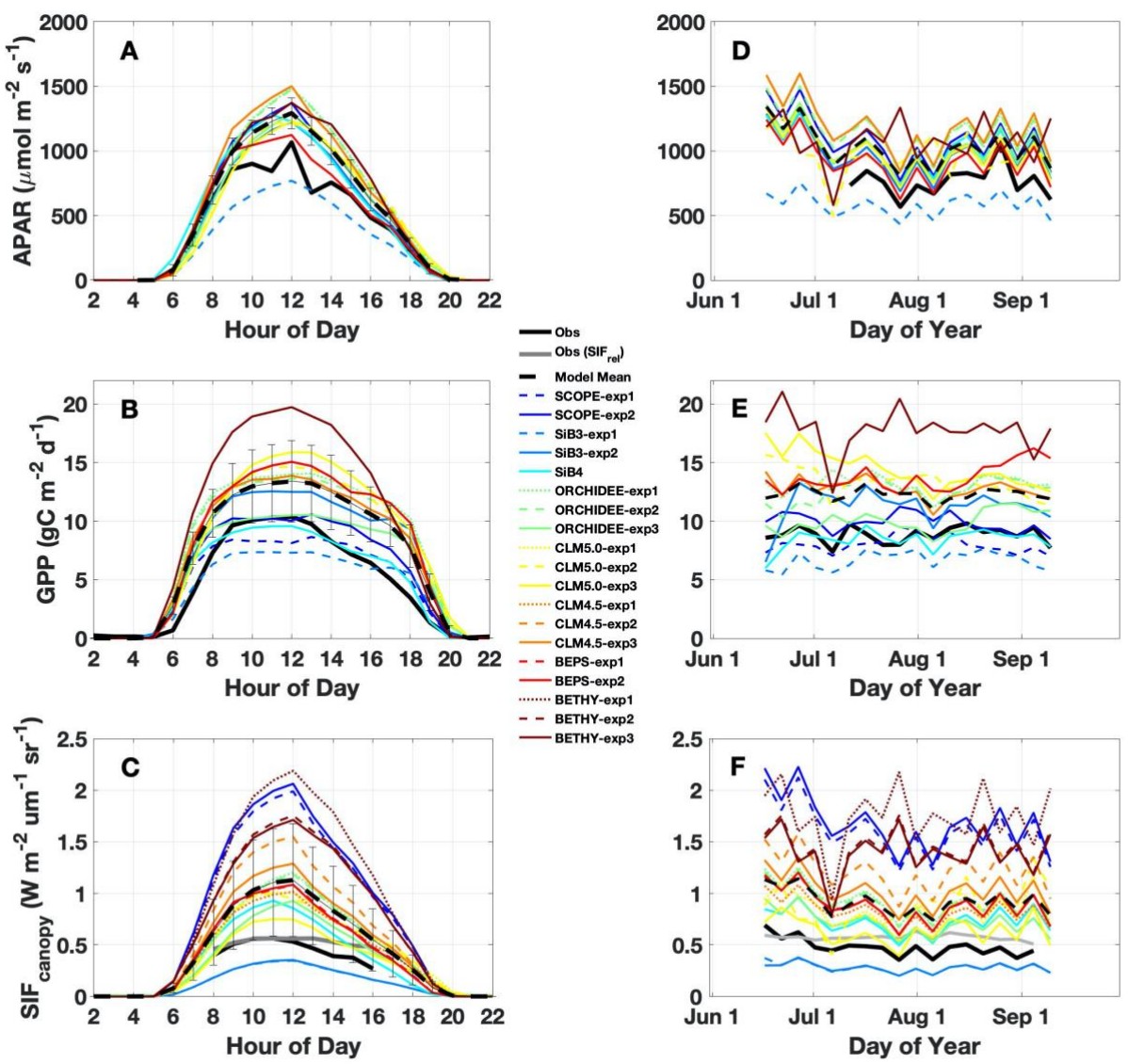


**Figure 2**. Observed and simulated diurnal and synoptic cycles of APAR, GPP and SIF. Diurnal cycles
(A-C) are averaged over July-August, 2017. Synoptic cycles (D-F) are plotted as 5-day averages
from June 15 – Sep 15. Observations are shown in black, with relative SIF ($SIF_{canopy}$ / far red
reflected radiance) included in (C, F) in grey. The across model average (dashed black) represents
the average of "best-case" model scenarios (solid lines; SCOPE-exp2, SiB3-exp2, SiB4, ORCHIDEE-
exp3, CLM5.0-exp3, CLM4.5-exp3, BEPS-exp2, BETHY-exp3) with uncertainty bars indicating the
across model 1 sigma uncertainty.

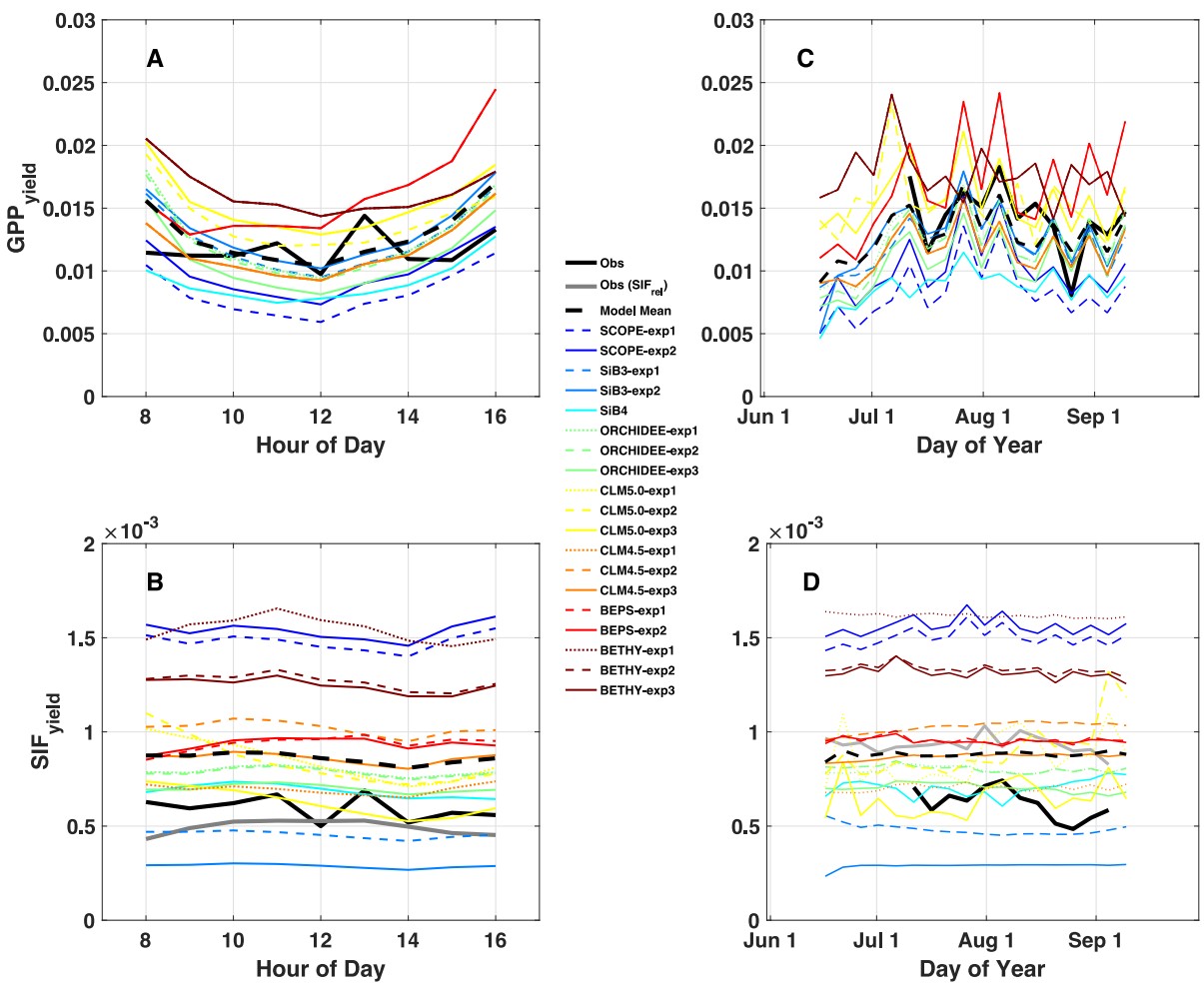


**Figure 3**. Same as Figure 2 except for $SIF_{yield}$ and $GPP_{yield}$. Here, $SIF_{yield} = SIF_{canopy}$ / APAR, and
$GPP_{yield}$ = GPP / APAR. As with Figure 2, the left column shows the mean diurnal cycle, and the
right column shows a time series of 5-day averages.

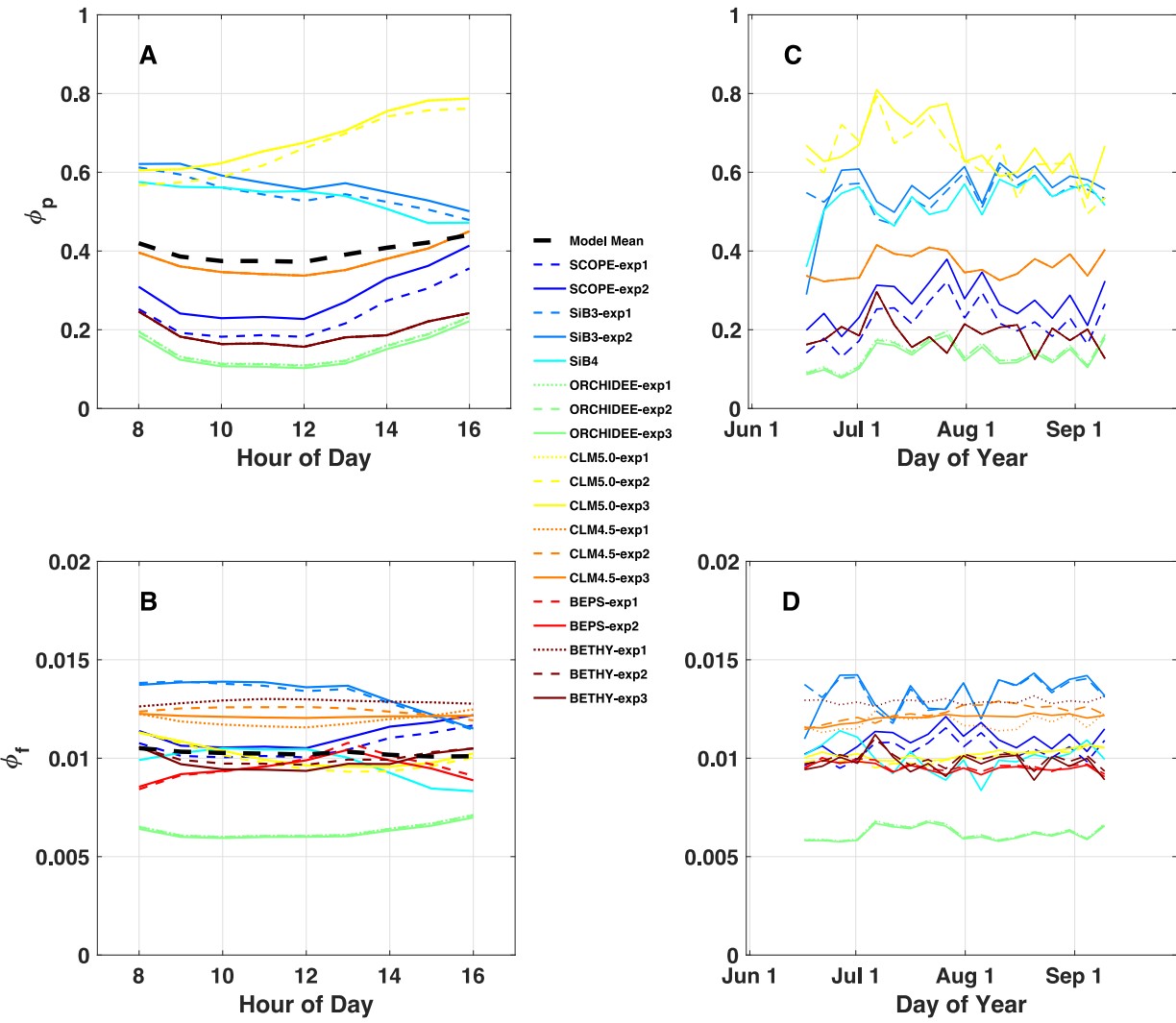


**Figure 4**. Same as Figure 2, except for quantum yield of fluorescence ($\phi_F$) and photochemistry
($\phi_P$).



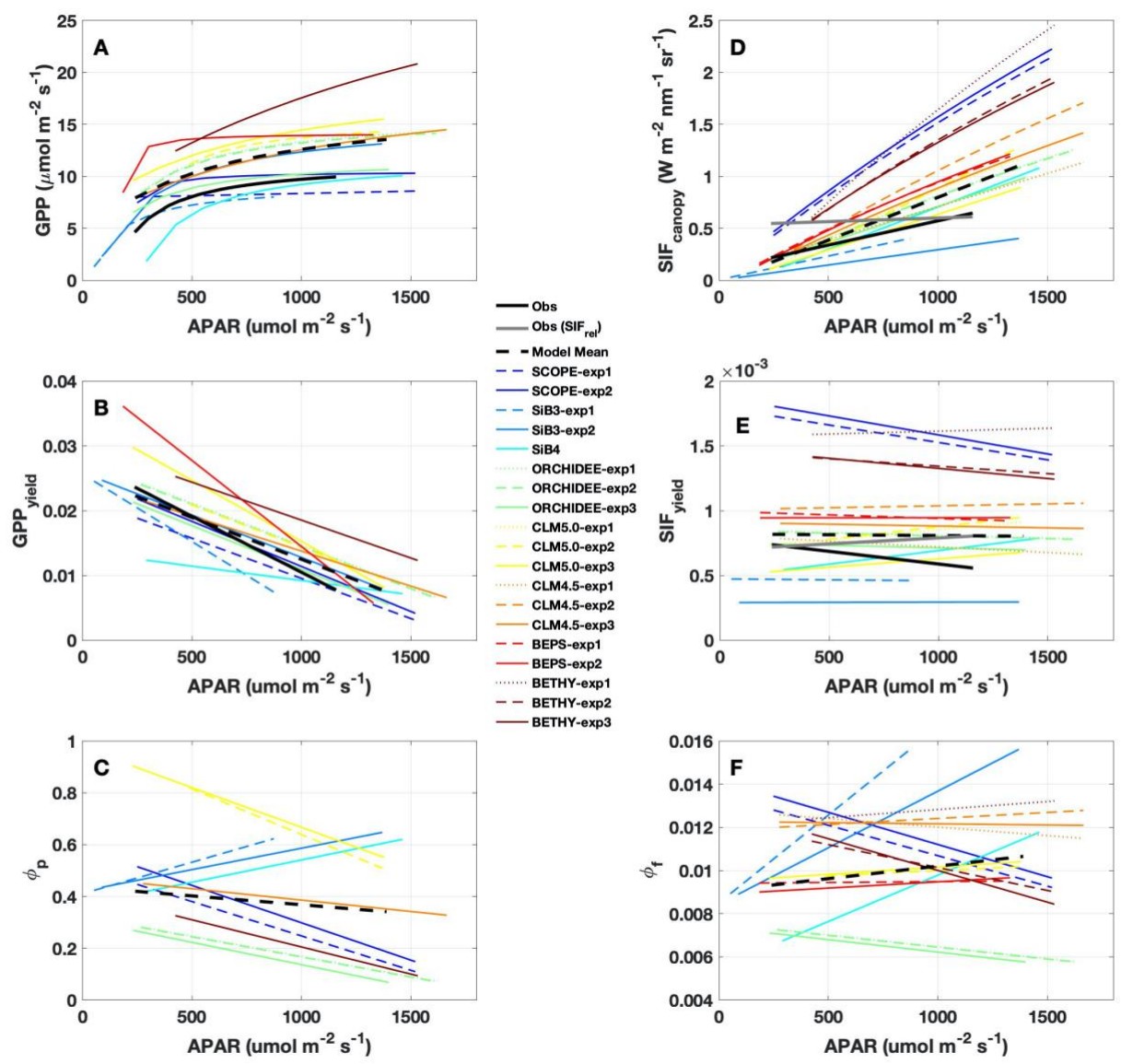


**Figure 5**. Observed and predicted change in GPP, SIF, and yields with APAR.  Regression lines are
shown for (A) GPP, (B) GPP$_{yield}$, (C) photochemical quantum yield ($\phi_p$), (D) SIF$_{canopy}$, (E) SIF$_{yield}$, (F)
fluorescence quantum yield ($\phi_f$), as a function of APAR, using daily mean (8 am – 4 pm local)
values over the period July-August 2017. Observations are shown in solid black, individual models
and experiments in color, the across model average in dashed black. Relative SIF is shown in grey
in (D) and (E).

**Tables**

| Model (citation) | Model Experiments | Stomatal Conductance | Canopy Type / Radiation | Stress | Vcmax | LAI | $k_N$ | Leaf-to-Canopy Scaling | Parameter Optimization |
|---|---|---|---|---|---|---|---|---|---|
| SCOPE v1.73 (van der Tol, 2014) | SCOPE-exp1 <br><br> SCOPE-exp2 | Ball-Berry-Woodrow | Multi-layer Sunlit/Shaded = Yes <br><br> Fpar/APAR = semi-analytical canopy radiative model (based on SAIL) | Ta stress | Prescribed (30) <br><br> Seasonally calibrated to NR1 | Prescribed (4.0 m² m⁻²) | Adapted to drought stressed Mediterranean species including high temperature correction (Tol et al., 2014; Flexas et al., 2002) | 60 layer 1D radiative transfer | Hand-tuned to NR1 (Raczka et al., 2016) |
| BETHY (Norton et al., 2019) | BETHY-exp1 <br><br> BETHY-exp2 <br><br> BETHY-exp3 | Ball-Berry-Woodrow | Same as SCOPE | Ta stress | Prior is a function of Ta | Prescribed (4.0 m² m⁻²) | Adapted to unstressed cotton species (Tol et al., 2014) <br><br> Adapted to drought stressed Mediterranean species including high temperature correction (Tol et al., 2014; Flexas et al., 2002) <br><br> Adapted to drought stressed Mediterranean species (Flexas et al., 2002) | SCOPE radiative transfer. f(Ta, APAR, structure, leaf composition) via dependence of photosynthetic rate on φ_F | Default |
| ORCHIDEE (Bacour et al., 2019) | ORCHIDEE-exp1 <br><br> ORCHIDEE-exp2 <br><br> ORCHIDEE-exp3 | Yin-Struik | Big Leaf Sunlit/Shaded = No <br><br> APAR = Beer-Lambert law depending on LAI and extinction factor = 0.5 | Ta stress <br><br> Ta and water stress (Yin and Struik, 2009) <br><br> Same as exp 1 | f (leaf age, CO₂, Ta, water stress) | Prognostic | Adapted to needleleaf species (Porcar-Castell et al., 2011) and unstressed Mediterranean species (Flexas, 2002), with added dependence on PAR, temperature, and φ_P | Parametric representation of SCOPE (v1.61) to emulate radiative transfer within canopy for PSI/II. | Default <br><br> Default <br><br> Global ENF PFT optimized against OCO-2 |
| BEPS (Qiu et al., 2019) | BEPS-exp1 <br><br> BEPS-exp2 | Leuning | Two Leaf Sunlit/Shaded = Yes <br><br> Fpar = semi-analytical canopy radiative transfer | Soil water stress factor (ratio of measured soil available water to maximum plant available water) | Prescribed | Prescribed (4.0 m² m⁻²) | Adapted to water stressed Mediterranean species (Galmes et al., 2007) <br><br> Adapted to drought stressed Mediterranean species including high temperature correction (Tol et al., 2014; Flexas et al., 2002) | Parametric representation of radiative transfer physics to account for canopy scattering effects | Default |
| CLM4.5 (Raczka et al., 2019) | CLM4.5-exp1 <br><br> CLM4.5-exp2 <br><br> CLM4.5-exp3 | Ball-Berry-Woodrow | Two Big Leaf Sunlit/Shaded = Yes | Ta(Vcmax); soil moisture stress uses Btran parameterization (function of column rooting profile and soil water potential) | Prescribed (calibrated against observed GPP at NR1) | Prescribed (4.0 m² m⁻²) | Adapted to water stressed Mediterranean species (Galmes et al., 2007) <br><br> Adapted to needleleaf species (Porcar-Castell et al., 2011); Accounts for sustained NPQ (k_S) separately from reversible NPQ (k_R). k_S is calibrated to NR1 Tair. k_R is fixed in time <br><br> same as Exp 2, but k_R is seasonal | $\kappa_{740}$ = f(Vcmax, SZA), calibrated to offline SCOPE runs using prescribed canopy characteristics at NR1 | Hand-tuned to NR1 (Raczka et al., 2016) |
| CLM5.0 (unpublished) | CLM5.0-exp1 <br><br> CLM5.0-exp2 <br><br> CLM5.0-exp3 | Medlyn | Two Big Leaf Sunlit/Shaded = Yes | Plant hydraulic water stress (Sperry and Love, 2015; Lawrence et al., 2019) accounting for water demand and supply | f (soil moisture, nitrogen), calibrated to NR1 | Prescribed (4.0 m² m⁻²) | Adapted to water stressed Mediterranean species (Galmes et al., 2007) | $\kappa_{740}$ = f(Vcmax), calibrated to offline SCOPE runs from Lee et al. (2015) <br><br> $\kappa_{740}$ = f(Vcmax, SZA), calibrated to offline SCOPE runs w/ prescribed canopy characteristics at NR1 <br><br> Escape ratio (f_esc), derived from NIRv and fPAR (Zeng et al., 2019) | Default |
| SiB3 (Baker et al., 2003, 2008) SiB4 (Haynes et al., 2019a,b) | SiB3-exp1 <br><br> SiB3-exp2 <br><br> SiB4 | Ball-Berry-Woodrow | Big Leaf Sunlit/Shaded = No | Downregulation by VPD, Ta, and soil moisture | f (soil moisture) | Prescribed (MODIS) <br><br> Prescribed (4.0 m² m⁻²) <br><br> Prognostic | Adapted to drought stressed species (Tol et al., 2014) | $\kappa_{740}$ = f(Vcmax), calibrated to offline SCOPE runs from Lee et al. (2015) | Default |


**Table 1**. Summary of TBM-SIF models and within model experiments illustrating model
components that may have led to differences in modeled SIF., These include a representation of
stomatal-conductance (column 3), canopy absorption of incoming radiation (column 4), limiting
factors for photosynthesis (Stress, $V_{cmax}$, LAI; columns 5-7) and SIF ($k_N$; column 8), leaf-to-canopy
scaling of SIF (column 9), and parameter optimization (column 10).  The underlined model
experiment was used for model intercomparison .
