# Peer review of "Wide Discrepancies in the Magnitude and Direction of Modelled SIF in Response to Light"

_Biogeosciences, 2019_

## Referee Comment (RC1) · Georg Wohlfahrt (Referee) · 18 Feb 2020

General:

Parazoo et al. compare seven SIF-enabled TBMs against empirical SIF and GPP data from a subalpine evergreen coniferous forest. The models, which had SIF retro-fitted, share some common concepts but on the other hand differ widely in terms of other concepts, with corresponding impacts on simulated SIF. The authors describe the differences compared to the empirical data and discuss these in terms of the differences in model structure.

[Figure]

Interest in the adding SIF capabilities to TBMs is largely driven by the recent availability of global SIF satellite products which provides promising avenues for additional constraints on carbon cycling, especially for GPP. Given that this research field is still in its infancy, I think the scope of this study, even though limited to a single site and a few weeks of peak-vegetation period data, is justified. The manuscript is well written and I think the authors do a great job in navigating the reader through the complexity of the investigated TBMs without getting lost in the many aspects these models differ.

I have only really very few detailed comments (see below) and only one major comment, that is that I was wondering whether the model comparison would profit from adding simulations with the original SCOPE model. This model is some sort of golden standard for SIF modelling (in fact many of the investigated models have gleaned from SCOPE in one way or the other) and I could imagine that SCOPE simulations might provide a good benchmark for the investigated TBMs, which given their scope need to weigh complexity against realism. Even though SCOPE is much more complex in terms of the treatment of canopy radiative transfer and gas exchange, running it with pre-scribed meteo inputs and adjusting a few key parameters should be easy to do.

Detailed comments:

l. 60: and theoretical models suggest a non-linear response at leaf-scale (Gu et al. 2019)

l. 84: a needle is anatomically a leaf

l. 102: not so much at leaf-scale really

l. 103: the FLOX is missing in the list of tower-mounted spectrometer systems

Fig. 1: calling a 3-year average a climatology is a bit of a stretch in my view – maybe just refer to this as the 2015-2018 average?

l. 165-174: how representative are these measurements for the larger footprint of the flux tower?

l. 229: one sentence on the effects of complex terrain, for which NR1 is famous, on NEE and inferred GPP?

l. 260: wouldn't that be the Ball-Berry-Woodrow (BBW) model?

l. 261: and this simply the Leuning model?

Table 1: what is the difference between big-leaf and single layer models? Where do two-leaf big-leaf models fall into?

l. 573: sunlit/shaded leaf area fractions

l. 803-810: what are recommendations for model structure with respect to APAR?

l. 816: might refer to new approaches such as stomatal optimisation based on xylem hydraulics (Eller et al. 2020)

l. 821: here I would think we also need more data from a wider variety of plant species under in situ conditions, especially all kinds of stress, ideally combining active and passive chlorophyll fluorescence measurements

l. 833: for perspective - do the authors dare to say something about what they would expect from a similar model comparison for a well-watered high-LAI crop?

---

## Referee Comment (RC2) · Christiaan van der Tol (Referee) · 2 Mar 2020

This paper compares different process based terrestrial biosphere model (TBMs) that include solar induced chlorophyll fluorescence (SIF) as output. The models are briefly introduced, with emphasis on the different representations of SIF. The model output with respect to SIF and gross primary productivity (GPP) output is inter-compared, and comparisons are made to a time series of field measurements. The models diverged, and the authors relate the differences among the models to the underlying process descriptions: the estimates of APAR, energy partitioning in the leaf and radiative transfer of fluorescence.

The paper provides a good overview of current TBMs capable of simulating SIF. This is of interest to the readers. It has an informative title, abstract and figures. It does not introduce new concepts, but it compares existing model concepts and recommends strategies for improvement. The paper is well written and clear.

I have the following recommendations to consider in the preparation of the final manuscript (all minor):

1. Make the paper (even) more inviting for readers who are unfamiliar with the terminology of SIF. In Line 208 $SIF_{yield}$ is first used, later in lines 593-602, it is defined, and the difference with $SIF_{rel}$ is discussed. It may be helpful to introduce $SIF_{yield}$, $SIF_{rel}$ and $\Phi_F$ together and earlier, explaining why these three are used for comparison in this paper (in Figs 3 and 4), and what they mean.

2. Lines 623-626. I did not grasp the following reasoning: 'Finally, we note that PhotoSpec scans of leaf-level emissions are averaged and reported here as canopy averages, while model output is reported at the top of the canopy, which accounts for within-canopy radiative transfer, re-absorption of SIF, and shaded canopies, causing lower emissions compared to the canopy average.' Aren't the top-of-canopy measurements also affected by within-canopy radiative transfer etcetera?

3. Continuation of previous point: The difference between the measurements and the simulations is that the measurements are the average of small footprints at multiple viewing angles, whereas the models are nadir values, as explained in the 'apples to apples' section (line 691). I presume the radiative transfer factor $\kappa_{740}$ was derived from SCOPE simulations in nadir. With SCOPE it is possible to estimate $\kappa_{740}(\theta_o)$ for multiple observation angles, and then take the average. Thus it is possible to compare apples to apples. I understand the TBM's do not have this right now, but at least I would have expected that to be part of the discussion, or as part of recommendation 5, which now only mentions instruments with a wider FOV.

4. Line 566, Strictly, x is not the fraction of absorbed light not used in photosynthesis, if

this refers to the variable 'x' in the model of Lee et al. and Van der Tol, because when x = 0, this fraction is 0.17 due to constitutive heat dissipation.

5. Line 728-730. 'The fact that relative SIF is the least sensitive [] reduces the sensitivity to APAR and reveals a strong SIF response to changes in photochemical quenching'. Yes, that seems to be the case, but perhaps a few lines can be added to guide the reader through this argument (see also point 1).

6. Line 811, recommendation 2. Is it the water stress formulation, or the parameter values, i.e. the values for the Ball-Berry parameters?

7. In Line 680, there is a reference to Figure 6, which is not in the manuscript

8. Figure 3C and 3D. What is the temporal resolution of these data? Multiple-day averages? It takes some effort to relate the spikes to the wet and dry periods described in the text.

Technical comments

Line 290, sentence starting 'The quantum yield' has an extra 'to'

Line 365 and elsewhere, I recommend to spell out 'met forcing'

Line 508, 'eaves' should be 'leaves'

Figures S1 and S4 are reversed

The labels in Figure S7 are too small

The legend in Figure S8 is too small

---

## Author Comment (AC1) · 7 Apr 2020

Reviewer:

General: Parazoo et al. compare seven SIF-enabled TBMs against empirical SIF and GPP data from a subalpine evergreen coniferous forest. The models, which had SIF retro-fitted, share some common concepts but on the other hand differ widely in terms of other concepts, with corresponding impacts on simulated SIF. The authors describe the differences compared to the empirical data and discuss these in terms of the differences in model structure.

Interest in the adding SIF capabilities to TBMs is largely driven by the recent availability of global SIF satellite products which provides promising avenues for additional constraints on carbon cycling, especially for GPP. Given that this research field is still in its infancy, I think the scope of this study, even though limited to a single site and a few weeks of peak-vegetation period data, is justified. The manuscript is well written and I think the authors do a great job in navigating the reader through the complexity of the investigated TBMs without getting lost in the many aspects these models differ.

Author:

We thank the reviewer for the nice feedback and helpful comments, and for appreciating our decision to keep our scope of study limited. Our hope is to build off the baseline findings reported here.

Reviewer:

I have only really very few detailed comments (see below) and only one major comment, that is that I was wondering whether the model comparison would profit from adding simulations with the original SCOPE model. This model is some sort of golden standard for SIF modelling (in fact many of the investigated models have gleaned from SCOPE in one way or the other) and I could imagine that SCOPE simulations might provide a good benchmark for the investigated TBMs, which given their scope need to weigh complexity against realism. Even though SCOPE is much more complex in terms of the treatment of canopy radiative transfer and gas exchange, running it with pre-scribed meteo inputs and adjusting a few key parameters should be easy to do.

Author:

This was a great recommendation and worth the small amount of extra work. We now include results from SCOPE v1.73 with prescribed met input for the year of study (2017) and vegetation parameters (LAI, canopy height, leaf chlorophyll content, and Vcmax) calibrated to NR1 according to Raczka et al., 2019. Results from the stand-alone

version of SCOPE are quite similarly qualitatively and quantitatively to the coupled version with BETHY (high bias in APAR and SIF), except with improved diurnal and synoptic variability compared to PhotoSpec. This provides a nice benchmark for TBM-SIFs in this study. We provide a description of SCOPE in the methods, references to SCOPE results throughout, and plots of SCOPE in all relevant figures (inc Figs 2-5 in the main text).

Reviewer:

Detailed comments:

l. 60: and theoretical models suggest a non-linear response at leaf-scale (Gu et al. 2019)

Author:

Corrected as follows:

"Spaceborne data indicate a linear relationship between SIF and GPP at large spatial (kilometer) and temporal (bi-weekly) scales (e.g., Sun et al., 2017) for several ecosystems, while theoretical models and ground-based measurements indicate a more non-linear relationship at leaf and canopy scales (Zhang et al., 2016; Gu et al., 2019; van der Tol et al., 2014; Magney et al., 2017, 2019a)"

Reviewer:

l. 84: a needle is anatomically a leaf

Author:

Changed 'needle/leaf' to 'leaf'

Reviewer:

l. 102: not so much at leaf-scale really

Author:

Changed 'leaf to canopy scale' to 'canopy scale'

Reviewer:

l. 103: the FLOX is missing in the list of tower-mounted spectrometer systems

Author:

We added FLOX and reference to Shan et al., 2019 and Julitta et al., 2017

Shan, N., Ju, W., Migliavacca, M., Martini, D., Guanter, L., Chen, J., Goulas, Y., Zhang, Y.: Modeling canopy conductance and transpiration from solar‐induced chlorophyll fluorescence. Agricultural and Forest Meteorology, 268, 189–201, 2019.

Julitta, T., Burkart, A., Colombo, R., Rossini, M., Schickling, A., Migliavacca, M., Cogliati, S., Wutzler, T., Rascher, U.: Accurate measurements of fluorescence in the O2A and O2B band using the FloX spectroscopy system - results and prospects. In: Proc. Potsdam GHG Flux Workshop: From Photosystems to Ecosystems, 24–26 October 2017, Potsdam, Germany. https://www.potsdam-flux-workshop.eu/, 2017

Reviewer:

Fig. 1: calling a 3-year average a climatology is a bit of a stretch in my view – maybe just refer to this as the 2015-2018 average?

Author:

Yes, thank you

Reviewer:

l. 165-174: how representative are these measurements for the larger footprint of the flux tower?

Author:

The answer is not very. We added the following stipulation at the end of the paragraph:

"We note that APAR measurements are only as representative as the distribution of PAR sensors beneath the canopy; while they are placed within the footprint of SIF (Sec 2.2.3) and fetch of eddy covariance (Sec 2.2.4) measurements, they cannot be a perfect representation of canopy APAR for each eddy covariance and SIF measurement."

Reviewer:

l. 229: one sentence on the effects of complex terrain, for which NR1 is famous, on NEE and inferred GPP?

Author:

Good point. The location does not have a significant impact on daytime fluxes, but we added the following sentence for full disclosure.

"We note the tower location near the Continental Divide in the Rocky Mountains of Colorado does present slope flow challenges for eddy covariance during nighttime, but the relatively flat area of the tower reduces impact on daytime flux measurements (Burns et al., 2018)."

Burns, S. P., Swenson, S. C., Wieder, W. R., Lawrence, D. M., Bonan, G. B., Knowles, J. F., and Blanken, P. D.: A comparison of the diel cycle of modeled and measured latent heat flux during the warm season in a Colorado subalpine forest, Journal of Advances in Modeling Earth Systems, 10, 617–651, 2018.

Reviewer:

l. 260: wouldn't that be the Ball-Berry-Woodrow (BBW) model? l. 261: and this simply the Leuning model?

Author:

Corrected in Sec 2.3.2 and in Table 1

Reviewer:

Table 1: what is the difference between big-leaf and single layer models? Where do two-leaf big-leaf models fall into?

Author:

Thank you for pointing out these differences. The models can be classified as follows.

BETHY = multiple layers (sunlit/shaded) ORCHIDEE/SIB3/4 = big leaf (sunlit only) CLM4.5/5 = two big leaf (sunlit/shaded) BEPS = two leaf (sunlit/shaded)

We clarify these differences in Table 1 and in Section 2.3.1 as shown below

"These differences, which are summarized in Table 1, include the representation of stomatal-conductance (all use Ball-Berry except CLM5.0, BEPS, and ORCHIDEE), canopy absorption of incoming radiation (all account for sunlit/shaded radiation except ORCHIDEE, SIB3, and SIB4), limiting factors for photosynthesis (Vcmax, LAI, radiation, stress) and SIF (kN, fluorescence photon re-absorption), scaling and radiative transfer methods for transferring leaf-level SIF simulations to top of canopy, and parameter optimization."

Reviewer:

l. 573: sunlit/shaded leaf area fractions

Author:

corrected, thank you

Reviewer:

l. 803-810: what are recommendations for model structure with respect to APAR?

Author:

We added the following recommendation at the end of Area 1, keeping in mind the stipulation that there is really no perfect in situ APAR measurement:

"We recommend further site-level investigation of observed and simulated canopy light absorption, emphasizing comparison of multi-layer and multi-leaf radiation schemes accounting for sunlit and shaded leaf area."

Reviewer:

l. 816: might refer to new approaches such as stomatal optimisation based on xylem hydraulics (Eller et al. 2020)

Author:

Agreed. We added the following recommendation at the end of Area 2:

"We also recommend more inclusion of stomatal optimization models (e.g., Eller et al., 2020) as optional parameterizations for TBMs, to better account for plant hydraulic functioning under water stress compared to the more widely used semi-empirical models."

l. 821: here I would think we also need more data from a wider variety of plant species under in situ conditions, especially all kinds of stress, ideally combining active and passive chlorophyll fluorescence measurements

Agreed. We added the following recommendation at the end of Area 3: "We also emphasize a need for more simultaneous measurements of active and passive chlorophyll fluorescence to determine the temporal dynamics of competing pathways (PQ, NPQ) from a wider variety of plant species under ambient conditions and different levels of stress."

Reviewer:

l. 833: for perspective - do the authors dare to say something about what they would expect from a similar model comparison for a well-watered high-LAI crop?

Author:

We added a 6th bullet point at the end:

"Finally, we note that our focus on a water limited subalpine evergreen needleleaf forest represents a challenging case study for models and observations. In many cases, there is strong covariance between LAI, SIF, APAR and GPP in cropping systems (Dechant et al., 2020), but because this study site experiences little change in canopy structure and APAR throughout the season (Magney et al, 2019b), our study sought to provide more explicit insight into the models sensitivity to photosynthesis and fluorescence. As such, it is possible that we would see more convergence of results, and a reduction in confounding effects (e.g., decreased NPQ), in a well-watered high-LAI cropping system. We therefore recommend similar model-observation assessments across a wider range of biota and climate."

––––––––––––––––––––

---

## Author Comment (AC2) · 7 Apr 2020

Reviewer

This paper compares different process based terrestrial biosphere model (TBMs) that include solar induced chlorophyll fluorescence (SIF) as output. The models are briefly introduced, with emphasis on the different representations of SIF. The model output with respect to SIF and gross primary productivity (GPP) output is inter-compared, and comparisons are made to a time series of field measurements. The models diverged, and the authors relate the differences among the models to the underlying process descriptions: the estimates of APAR, energy partitioning in the leaf and radiative transfer

of fluorescence.

The paper provides a good overview of current TBMs capable of simulating SIF. This is of interest to the readers. It has an informative title, abstract and figures. It does not introduce new concepts, but it compares existing model concepts and recommends strategies for improvement. The paper is well written and clear. I have the following recommendations to consider in the preparation of the final manuscript (all minor):

Author

Thank you for the very kind and helpful review.

Reviewer

1. Make the paper (even) more inviting for readers who are unfamiliar with the terminology of SIF. In Line 208, SIFyield is first used, later in lines 593-602, it is defined, and the difference with SIFrel is discussed. It may be helpful to introduce SIFyield, SIFrel and phi_F together and earlier, explaining why these three are used for comparison in this paper (in Figs 3 and 4), and what they mean.

Author

We thank the reviewer for this helpful suggestion. We added a new section toward the beginning of the methods to clarify these differences, merging information from line 208 and 593-602.

"2.2.2 SIF Yield

We define and clarify three important quantities that define the relationship between absorbed light and emitted SIF at leaf and canopy scales. ΦT_F is the quantum yield of fluorescence, representing the probability an absorbed photon will be fluoresced. This quantity can be observed at leaf level using PAM fluorimetry, or calculated by models as a function of rate coefficients for energy transfer (Sec 2.3.3). SIFyield is the canopy emitted SIF per photon absorbed. The quantify is estimated from models and observations as the ratio of absolute canopy SIF and APAR (SIFcanopy/APAR). SIFyield is our best attempt to account for the effect of (a) canopy absorbed light and (b) SIF re-absoprtion within the canopy on the canopy integrated emission of SIF. However, factors such as observation angle, fraction of sunit/shaded canopy components, and difference in footprint from APAR, necessitates an additional diagnostic variable defined as relative SIF (SIFrel). SIFrel is emitted SIF per reflected radiance in the far red spectrum where SIF retrievals occur (SIF/Reffr). This is useful because is normalizes for the exact amount of 'illuminated' canopy elements within the sensor field of view, whereas APAR measurements are integrated for the entire canopy.

These quantities represent different but equally important versions of reality. It is difficult for models to exactly reproduce the distribution and timing of sunlight in the canopy as observed by PhotoSpec. While SIFrel removes model-observations differences in illumination, it confounds our interpretation of the relationship with GPPyield, which is derived from APAR. As such, we provide both results to be comprehensive, but note the temporal stability associated with SIFrel as the more physical interpretation of canopy yield for this short period of study."

Reviewer

2. Lines 623-626. I did not grasp the following reasoning: 'Finally, we note that PhotoSpec scans of leaf-level emissions are averaged and reported here as canopy averages, while model output is reported at the top of the canopy, which accounts for within-canopy radiative transfer, re-absorption of SIF, and shaded canopies, causing lower emissions compared to the canopy average.' Aren't the top-of-canopy measurements also affected by within-canopy radiative transfer etcetera?

Author

Thank you for pointing out this source of confusion. We tried to clarify as follows:

"Finally, we clarify an important difference between observed and predicted estimates

of canopy average SIF. PhotoSpec scans direct emissions from sunlit and shaded leaves within the canopy, thus observing the 'total' emission from leaves in the instrument FOV. We then average each of these leaf-level scans and report as canopy averages. Model output, in contrast, is reported at the TOC, which represents the 'net' emission from leaves after attenuation in the canopy (through canopy radiative transfer, re-absorption of SIF, and shading). Assuming sunlit and shaded leaves within the canopy emit at the same rate as TOC leaves, attenuation will reduce the effective signal from leaf-level emissions within the canopy. As such, the average of leaf level emissions (canopy average) is expected to be lower than the net emission of leaves reaching the top of canopy. This is important because CLM4.5 shows strong attenuation of SIF from leaf-level to TOC, decreasing by a factor of 2-3 at midday (Fig S7). The interpretation here is that the model bias in absolute SIF may actually be higher than reported here; however, we note that more quantitative information on the observed fraction of sunlit vs shaded leaves and comparative top-of-canopy SIF values for the same canopy elements are needed (to account for off-nadir SIF viewing) for more accurate determination of scaling between observed canopy and top-of-canopy SIF."

Reviewer

3. Continuation of previous point: The difference between the measurements and the simulations is that the measurements are the average of small footprints at multiple viewing angles, whereas the models are nadir values, as explained in the 'apples to apples' section (line 691). I presume the radiative transfer factor _740 was derived from SCOPE simulations in nadir. With SCOPE it is possible to estimate _740 (_o) for multiple observation angles, and then take the average. Thus it is possible to compare apples to apples. I understand the TBM's do not have this right now, but at least I would have expected that to be part of the discussion, or as part of recommendation 5, which now only mentions instruments with a wider FOV.

Author

Very excellent point. We added the following sentence to recommendation 5

"More effort is also needed to better align models with observations, for example by leveraging three-dimensional capabilities in SCOPE (and other RTMs) to directly account for multiple observation angles."

Reviewer

4. Line 566, Strictly, x is not the fraction of absorbed light not used in photosynthesis, if this refers to the variable 'x' in the model of Lee et al. and Van der Tol, because when x = 0, this fraction is 0.17 due to constitutive heat dissipation.

Author

Thank you for clarifying. We removed the statement that x refers to the "fraction of absorbed light not used in photosynthesis"

Reviewer

5. Line 728-730. 'The fact that relative SIF is the least sensitive [] reduces the sensitivity to APAR and reveals a strong SIF response to changes in photochemical quenching'. Yes, that seems to be the case, but perhaps a few lines can be added to guide the reader through this argument (see also point 1).

Author

We agree this is a difficult concept to grapple with. We try to clarify as follows:

"Our results indicate a wide range of SIF responses to APAR: TBM-SIFs and SCOPE are usually far too sensitive to APAR, observations of absolute SIF are less sensitive, and observations of relative SIF (SIFrel) are least sensitive (Fig. 5D). We remind the reader that SIFrel is normalized by the amount of far red light reflected from leaves in the FOV of PhotoSpec, and thus has reduced sensitivity to absorbed light than absolute SIF. The fact that SIFrel is the least sensitive to APAR means other processes are driving changes in SIF under increased light absorption. In this case, it reveals a

strong SIF response to changes in photochemical quenching."

Reviewer

6. Line 811, recommendation 2. Is it the water stress formulation, or the parameter values, i.e. the values for the Ball-Berry parameters?

Author

Here, we are referring to different kinds of the stomatal conductance models (ball-berry, leuning) and water stress (e.g., soil moisture scalar for attenuating conductance). We clarify

"The underlying photosynthetic models fail to simulate the magnitude of depression of observed GPP in the afternoon, regardless of how stomatal-conductance and water stress models and parameters are formulated"

Following Reviewer 1, we also advocate for more use of stomatal optimization models

"We also recommend more inclusion of stomatal optimization models (e.g., Eller et al., 2020) as optional parameterizations for TBMs, to better account for plant hydraulic functioning under water stress compared to the more widely used semi-empirical models."

Reviewer

7. In Line 680, there is a reference to Figure 6, which is not in the manuscript

Author

Good catch, we refer to Fig S8 now.

Reviewer

8. Figure 3C and 3D. What is the temporal resolution of these data? Multiple-day averages? It takes some effort to relate the spikes to the wet and dry periods described in the text.

Author

Thank you. We have clarified the temporal resolution in the text and figure caption.

Reviewer

Technical comments

Line 290, sentence starting 'The quantum yield' has an extra 'to': Line 365 and elsewhere, I recommend to spell out 'met forcing': Line 508, 'eaves' should be 'leaves': Figures S1 and S4 are reversed: The labels in Figure S7 are too small The legend in Figure S8 is too small

Author

All corrected

---

## Author Response (AR1)

Date: April 17, 2020
Subject: Cover Letter for Revised Submission of bg-2019-508
Dear Reviewers and Associate Editor,
We thank you all for taking time to provide thoughtful and constructive comments. We have
addressed all comments, and paid particular attention to (1) clarify new concepts such as
relative sif, (2) expand our concluding recommendations with more detailed strategies to
improve model formulation and model-observational analysis, and (3) benchmarking analysis
against stand along tower driven SCOPE simulations. The resulting manuscript is improved in
readability and outcomes.
Please find below our merged document containing comments from Reviewers 1 and 2 with
embedded Author Responses and Changes (Page 2-10) and Tracked Changes starting on Page
11. Note the line numbers in the Reviewer comments (black font) refer to our original
submission, while Page and line numbers in the Author Response (blue font) refer to the
"Tracked Changes" document below.
Best regards,
Dr Nicholas Parazoo (on behalf of all co-authors)
Jet  Propulsion Laboratory
4800 Oak Grove Drive
Mail Stop 200-233
Pasadena, CA 91109
Phone: 818.354.2973

Comments and Author Response to Reviewer 1:
General: Parazoo et al. compare seven SIF-enabled TBMs against empirical SIF and GPP data
from a subalpine evergreen coniferous forest. The models, which had SIF retro-fitted, share
some common concepts but on the other hand differ widely in terms of other concepts, with
corresponding impacts on simulated SIF. The authors describe the differences compared to the
empirical data and discuss these in terms of the differences in model structure.
Interest in the adding SIF capabilities to TBMs is largely driven by the recent availability of
global SIF satellite products which provides promising avenues for additional constraints on
carbon cycling, especially for GPP. Given that this research field is still in its infancy, I think the
scope of this study, even though limited to a single site and a few weeks of peak-vegetation
period data, is justified. The manuscript is well written and I think the authors do a great job in
navigating the reader through the complexity of the investigated TBMs without getting lost in
the many aspects these models differ.
We thank the reviewer for the nice feedback and helpful comments, and for appreciating our
decision to keep our scope of study limited. Our hope is to build off the baseline findings
reported here.
I have only really very few detailed comments (see below) and only one major comment, that is
that I was wondering whether the model comparison would profit from adding simulations with
the original SCOPE model. This model is some sort of golden standard for SIF modelling (in fact
many of the investigated models have gleaned from SCOPE in one way or the other) and I could
imagine that SCOPE simulations might provide a good benchmark for the investigated TBMs,
which given their scope need to weigh complexity against realism. Even though SCOPE is much
more complex in terms of the treatment of canopy radiative transfer and gas exchange, running
it with pre-scribed meteo inputs and adjusting a few key parameters should be easy to do.
This was a great recommendation and worth the extra effort. We now include results from
SCOPE v1.73 with prescribed met input for the year of study (2017) and vegetation parameters
(LAI, canopy height, leaf chlorophyll content, and Vcmax) calibrated to NR1 according to Raczka
et al., 2019. Results from the stand-alone version of SCOPE are quite similarly qualitatively and
quantitatively to the coupled version with BETHY (high bias in APAR and SIF), except with
improved diurnal and synoptic variability compared to PhotoSpec. This provides a nice
benchmark for TBM-SIFs in this study.  We provide a description of SCOPE in the methods,
references to SCOPE results throughout, and plots of SCOPE in all relevant figures (including
Figs 2-5 in the main text).
Detailed comments:
l. 60: and theoretical models suggest a non-linear response at leaf-scale (Gu et al. 2019)
Statement added as follows:

"Spaceborne data indicate a linear relationship between SIF and GPP at large spatial (kilometer) and temporal (bi-weekly) scales (e.g., Sun et al., 2017) for several ecosystems, while theoretical models and ground-based measurements indicate a more non-linear relationship at leaf and canopy scales (Zhang et al., 2016; Gu et al., 2019; van der Tol et al., 2014; Magney et al., 2017, 2019a).

l. 84: a needle is anatomically a leaf

Changed 'needle/leaf' to 'leaf'

l. 102: not so much at leaf-scale really

Changed 'leaf to canopy scale' to 'canopy scale'

l. 103: the FLOX is missing in the list of tower-mounted spectrometer systems added FLOX and reference to Shan et al., 2019 and Julitta et al., 2017

Shan, N., Ju, W., Migliavacca, M., Martini, D., Guanter, L., Chen, J., Goulas, Y., Zhang, Y.:
    Modeling canopy conductance and transpiration from solar-induced chlorophyll
    fluorescence. Agricultural and Forest Meteorology, 268, 189–201, 2019.

Julitta, T., Burkart, A., Colombo, R., Rossini, M., Schickling, A., Migliavacca, M., Cogliati, S.,
    Wutzler, T., Rascher, U.: Accurate measurements of fluorescence in the O2A and O2B band
    using the FloX spectroscopy system - results and prospects. In: Proc. Potsdam GHG Flux
    Workshop: From Photosystems to Ecosystems, 24–26 October 2017, Potsdam, Germany.
    https://www.potsdam-flux-workshop.eu/, 2017

Fig. 1: calling a 3-year average a climatology is a bit of a stretch in my view – maybe just refer to this as the 2015-2018 average?

Yes, thank you l. 165-174: how representative are these measurements for the larger footprint of the flux tower?

Under most daytime conditions abd turbulent boundary layers, SIF measurements have a much smaller footprint compared to eddy covariance data, and thus are typically not representative of the larger ecosystem. We added the following stipulation at the end of the paragraph:

"We note that APAR measurements are only as representative as the distribution of PAR sensors beneath the canopy; while they are placed within the footprint of SIF (Sec 2.2.3) and fetch of eddy covariance (Sec 2.2.4) measurements, they cannot be a perfect representation of canopy APAR for each eddy covariance and SIF measurement."

l. 229: one sentence on the effects of complex terrain, for which NR1 is famous, on NEE and inferred GPP?

Good point. The location does not have a significant impact on daytime fluxes, but we added the following sentence for full disclosure.

"We note the tower location near the Continental Divide in the Rocky Mountains of Colorado does present slope flow challenges for eddy covariance during nighttime, but the relatively flat area of the tower reduces impact on daytime flux measurements (Burns et al., 2018)."

Burns, S. P., Swenson, S. C., Wieder, W. R., Lawrence, D. M., Bonan, G. B., Knowles, J. F., and Blanken, P. D.: A comparison of the diel cycle of modeled and measured latent heat flux during the warm season in a Colorado subalpine forest, Journal of Advances in Modeling Earth Systems, 10, 617–651, 2018.

l. 260: wouldn't that be the Ball-Berry-Woodrow (BBW) model?
l. 261: and this simply the Leuning model?

Corrected in Sec 2.3.2 and in Table 1

Table 1: what is the difference between big-leaf and single layer models? Where do two-leaf big-leaf models fall into?

Thank you for pointing out these differences. The models can be classified as follows.

BETHY = multiple layers (sunlit/shaded)
ORCHIDEE/SIB3/4 = big leaf (sunlit only)
CLM4.5/5 = two big leaf (sunlit/shaded)
BEPS = two leaf (sunlit/shaded)

We clarify these differences in Table 1 and in Section 2.3.1 as shown below

"These differences, which are summarized in Table 1, include the representation of stomatal-conductance (all use Ball-Berry except CLM5.0, BEPS, and ORCHIDEE), canopy absorption of incoming radiation (all account for sunlit/shaded radiation except ORCHIDEE, SIB3, and SIB4), limiting factors for photosynthesis (Vcmax, LAI, radiation, stress) and SIF ($k_N$, fluorescence photon re-absorption), scaling and radiative transfer methods for transferring leaf-level SIF simulations to top of canopy, and parameter optimization."

l. 573: sunlit/shaded leaf area fractions corrected, thank you l. 803-810: what are recommendations for model structure with respect to APAR?

We added the following recommendation at the end of Area 1 of Section 5, keeping in mind the stipulation that there is really no perfect in situ APAR measurement:

"We recommend further site-level investigation of observed and simulated canopy light absorption, emphasizing comparison of multi-layer and multi-leaf radiation schemes accounting for sunlit and shaded leaf area."

l. 816: might refer to new approaches such as stomatal optimisation based on xylem hydraulics (Eller et al. 2020)

Agreed. We added the following recommendation at the end of Area 2 of Section 5:

"We also recommend more inclusion of stomatal optimization models (e.g., Eller et al., 2020) as optional parameterizations for TBMs, to better account for plant hydraulic functioning under water stress compared to the more widely used semi-empirical models."

l. 821: here I would think we also need more data from a wider variety of plant species under in situ conditions, especially all kinds of stress, ideally combining active and passive chlorophyll fluorescence measurements

Agreed. We added the following recommendation at the end of Area 3 of Section 5:

"We also emphasize a need for more simultaneous measurements of active and passive chlorophyll fluorescence to determine the temporal dynamics of competing pathways (PQ, NPQ) from a wider variety of plant species under ambient conditions and different levels of stress."

l. 833: for perspective - do the authors dare to say something about what they would expect from a similar model comparison for a well-watered high-LAI crop?

We added a 6th bullet point at the end of Section 5:

"Finally, we note that our focus on a water limited subalpine evergreen needleleaf forest represents a challenging case study for models and observations. In many cases, there is strong covariance between LAI, SIF, APAR and GPP in cropping systems (Dechant et al., 2020), but because this study site experiences little change in canopy structure and APAR throughout the season (Magney et al, 2019b), our study sought to provide more explicit insight into the models sensitivity to photosynthesis and fluorescence. As such, it is possible that we would see more convergence of results, and a reduction in confounding effects (e.g., decreased NPQ), in a wellwatered high-LAI cropping system. We therefore recommend similar model-observation assessments across a wider range of biota and climate."

Comments and Author Response to Reviewer 1:

This paper compares different process based terrestrial biosphere model (TBMs) that include
solar induced chlorophyll fluorescence (SIF) as output. The models are briefly introduced, with
emphasis on the different representations of SIF. The model output with respect to SIF and
gross primary productivity (GPP) output is inter-compared, and comparisons are made to a time
series of field measurements. The models diverged, and the authors relate the differences
among the models to the underlying process descriptions: the estimates of APAR, energy
partitioning in the leaf and radiative transfer of fluorescence.

The paper provides a good overview of current TBMs capable of simulating SIF. This is of
interest to the readers. It has an informative title, abstract and figures. It does not introduce
new concepts, but it compares existing model concepts and recommends strategies for
improvement. The paper is well written and clear. I have the following recommendations to
consider in the preparation of the final manuscript (all minor):

Thank you for the very kind review.

1.  Make the paper (even) more inviting for readers who are unfamiliar with the terminology of
SIF. In Line 208, SIFyield is first used, later in lines 593-602, it is defined, and the difference
with SIFrel is discussed. It may be helpful to introduce SIFyield, SIFrel and phi_F together
and earlier, explaining why these three are used for comparison in this paper (in Figs 3 and
4), and what they mean.

We thank the reviewer for this helpful suggestion. We added a new section (2.2.2) toward the
beginning of the methods to clarify these differences, merging information from line 208 and
593-602.

"2.2.2 SIF Yield

We define and clarify three important quantities that define the relationship between absorbed
light and emitted SIF at leaf and canopy scales. $\phi_F$ is the quantum yield of fluorescence,
representing the probability an absorbed photon will be fluoresced. This quantity can be
observed at leaf level using PAM fluorimetry, or calculated by models as a function of rate
coefficients for energy transfer (Sec 2.3.3). $SIF_{yield}$ is the canopy emitted SIF per photon absorbed.
The quantify is estimated from models and observations as the ratio of absolute canopy SIF and
APAR ($SIF_{canopy}$/APAR). $SIF_{yield}$ is our best attempt to account for the effect of (a) canopy absorbed
light and (b) SIF re-absoprtion within the canopy on the canopy integrated emission of SIF.
However, factors such as observation angle, fraction of sunit/shaded canopy components, and
difference in footprint from APAR, necessitates an additional diagnostic variable defined as
relative SIF ($SIF_{rel}$). $SIF_{rel}$ is emitted SIF per reflected radiance in the far red spectrum where SIF
retrievals occur ($SIF/Ref_{fr}$). This is useful because is normalizes for the exact amount of
'illuminated' canopy elements within the sensor field of view, whereas APAR measurements are
integrated for the entire canopy.

These quantities represent different but equally important versions of reality. It is difficult for models to exactly reproduce the distribution and timing of sunlight in the canopy as observed by PhotoSpec. While $SIF_{rel}$ removes model-observations differences in illumination, it confounds our interpretation of the relationship with $GPP_{yield}$, which is derived from APAR. As such, we provide both results to be comprehensive, but note the temporal stability associated with $SIF_{rel}$ as the more physical interpretation of canopy yield for this short period of study."

2. Lines 623-626. I did not grasp the following reasoning: 'Finally, we note that PhotoSpec scans of leaf-level emissions are averaged and reported here as canopy averages, while model output is reported at the top of the canopy, which accounts for within-canopy radiative transfer, re-absorption of SIF, and shaded canopies, causing lower emissions compared to the canopy average.' Aren't the top-of-canopy measurements also affected by within-canopy radiative transfer etcetera?

Thank you for pointing out this source of confusion. We clarify as follows (Page 34, Line 4-19):

"Finally, we clarify an important difference between observed and predicted estimates of canopy average SIF. PhotoSpec scans direct emissions from sunlit and shaded leaves within the canopy, thus observing the 'total' emission from leaves in the instrument FOV. We then average each of these leaf-level scans and report as canopy averages. Model output, in contrast, is reported at the TOC, which represents the 'net' emission from leaves after attenuation in the canopy (through canopy radiative transfer, re-absorption of SIF, and shading). Assuming sunlit and shaded leaves within the canopy emit at the same rate as TOC leaves, attenuation will reduce the effective signal from leaf-level emissions within the canopy. As such, the average of leaf level emissions (canopy average) is expected to be lower than the net emission of leaves reaching the top of canopy.

This is important because CLM4.5 shows strong attenuation of SIF from leaf-level to TOC, decreasing by a factor of 2-3 at midday (Fig S7). The interpretation here is that the model bias in absolute SIF may actually be higher than reported here; however, we note that more quantitative information on the observed fraction of sunlit vs shaded leaves and comparative top-of-canopy SIF values for the same canopy elements are needed (to account for off-nadir SIF viewing) for more accurate determination of scaling between observed canopy and top-of-canopy SIF."

3. Continuation of previous point: The difference between the measurements and the simulations is that the measurements are the average of small footprints at multiple viewing angles, whereas the models are nadir values, as explained in the 'apples to apples' section (line 691). I presume the radiative transfer factor _740 was derived from SCOPE simulations in nadir. With SCOPE it is possible to estimate _740 (_o) for multiple observation angles, and then take the average. Thus it is possible to compare apples to apples. I understand the TBM's do not have this right now, but at least I would have expected that to be part of the discussion, or as part of recommendation 5, which now only mentions instruments with a wider FOV.

Very excellent point. We added the following sentence to area 5 of Section 5

"More effort is also needed to better align models with observations, for example by leveraging three-dimensional capabilities in SCOPE (and other RTMs) to directly account for multiple observation angles."

4. Line 566, Strictly, x is not the fraction of absorbed light not used in photosynthesis, if this refers to the variable 'x' in the model of Lee et al. and Van der Tol, because when x = 0, this fraction is 0.17 due to constitutive heat dissipation.

Thank you for clarifying. We removed the statement that x refers to the "fraction of absorbed light not used in photosynthesis"

5. Line 728-730. 'The fact that relative SIF is the least sensitive [] reduces the sensitivity to APAR and reveals a strong SIF response to changes in photochemical quenching'. Yes, that seems to be the case, but perhaps a few lines can be added to guide the reader through this argument (see also point 1).

We agree this is a difficult concept to grapple with. We try to clarify as follows (Page 38, Line 4-10):

"Our results indicate a wide range of SIF responses to APAR: TBM-SIFs and SCOPE are usually far too sensitive to APAR, observations of absolute SIF are less sensitive, and observations of relative SIF ($SIF_{rel}$) are least sensitive (Fig. 5D). We remind the reader that $SIF_{rel}$ is normalized by the amount of far red light reflected from leaves in the FOV of PhotoSpec, and thus has reduced sensitivity to absorbed light than absolute SIF. The fact that $SIF_{rel}$ is the least sensitive to APAR means other processes are driving changes in SIF under increased light absorption. In this case, it reveals a strong SIF response to changes in photochemical quenching."

6. Line 811, recommendation 2. Is it the water stress formulation, or the parameter values, i.e. the values for the Ball-Berry parameters?

Here, we are referring to different kinds of the stomatal conductance models (ball-berry, leuning) and water stress (e.g., soil moisture scalar for attenuating conductance). We clarify (Page 41, Line 7-9)

"The underlying photosynthetic models fail to simulate the magnitude of depression of observed GPP in the afternoon, regardless of how stomatal-conductance and water stress models and parameters are formulated"

Following Reviewer 1, we also advocate for more use of stomatal optimization models (Page 41, Line 13-16)

"We also recommend more inclusion of stomatal optimization models (e.g., Eller et al., 2020) as optional parameterizations for TBMs, to better account for plant hydraulic functioning under water stress compared to the more widely used semi-empirical models."

7. In Line 680, there is a reference to Figure 6, which is not in the manuscript

Good catch, we refer to Fig S8 now.

8. Figure 3C and 3D. What is the temporal resolution of these data? Multiple-day averages? It takes some effort to relate the spikes to the wet and dry periods described in the text.

Thank you. We have clarified the temporal resolution in the text and figure caption.

Technical comments

Line 290, sentence starting 'The quantum yield' has an extra 'to':
Line 365 and elsewhere, I recommend to spell out 'met forcing':
Line 508, 'eaves' should be 'leaves':
Figures S1 and S4 are reversed:
The labels in Figure S7 are too small
The legend in Figure S8 is too small

All corrected

[revised manuscript text omitted]

**Commented [NCP1]:** Dave/Sean –This is in response to R1 (Wolfahrt) regarding EC uncertainty in mountains. I stole this wording from the 2018 paper, please reword as needed.

[revised manuscript text omitted]

---

## Author Response (AR2)

Date: May 11, 2020
Subject: Cover Letter for Revised Submission of bg-2019-508
Dear Reviewers and Associate Editor,
We thank you for taking time to read through our revisions. We have addressed all minor
comments as discussed below. Please find below our merged document containing Author
Responses and Changes (Page 2) and Tracked Changes starting on Page 3.
Best regards,
Dr Nicholas Parazoo (on behalf of all co-authors)
Jet  Propulsion Laboratory
4800 Oak Grove Drive
Mail Stop 200-233
Pasadena, CA 91109
Phone: 818.354.2973

Comments and Author Response:

The authors have done a great job in replying to my comments and changing the manuscript accordingly. Especially, the addition of SCOPE model simulations benefits the analysis a lot, I find. I have just a few minor comments not requiring further reviewer assessment:

We agree the addition of SCOPE was extremely beneficial, thank you for the suggestion l. 45: "distribution of light across sunlit and shaded leaves" or "within-canopy distribution of direct and diffuse radiation"?!

We changed to the former, thanks

Fig. 1A: Replace "Climatology" in the legend

Done l. 807: show instead of shows

Done l. 887: "sunlit shaded fractions of leaf level" – what does that mean?

This is poor wording – we really just mean emission of sif from sunlit and shaded fractions of the leaf. We reworded the paragraph to clarify

[revised manuscript text omitted]